# Petal abscission is promoted by jasmonic acid-induced autophagy at Arabidopsis petal bases

Yuki Furuta [1,9], Haruka Yamamoto[1,9], Takeshi Hirakawa[1], Akira Uemura[1], Margaret Anne Pelayo [1,2], Hideaki Iimura[1,3], Naoya Katagiri[1], Noriko Takeda-Kamiya[4], Kie Kumaishi[5], Makoto Shirakawa[1,6], Sumie Ishiguro[7], Yasunori Ichihashi[4], Takamasa Suzuki[8], Tatsuaki Goh [1], Kiminori Toyooka [4], Toshiro Ito [1] ✉ & Nobutoshi Yamaguchi [1] ✉

In angiosperms, the transition from floral-organ maintenance to abscission determines reproductive success and seed dispersion. For petal abscission, cell-fate decisions specifically at the petal-cell base are more important than organ-level senescence or cell death in petals. However, how this transition is regulated remains unclear. Here, we identify a jasmonic acid (JA)-regulated chromatin-state switch at the base of Arabidopsis petals that directs local cell-fate determination via autophagy. During petal maintenance, co-repressors of JA signaling accumulate at the base of petals to block MYC activity, leading to lower levels of ROS. JA acts as an airborne signaling molecule transmitted from stamens to petals, accumulating primarily in petal bases to trigger chromatin remodeling. This allows MYC transcription factors to promote chromatin accessibility for downstream targets, including *NAC DOMAIN-CONTAINING PROTEIN102* (*ANAC102*). ANAC102 accumulates specifically at the petal base prior to abscission and triggers ROS accumulation and cell death via *AUTOPHAGY-RELATED GENE*s induction. Developmentally induced autophagy at the petal base causes maturation, vacuolar delivery, and breakdown of autophagosomes for terminal cell differentiation. Dynamic changes in vesicles and cytoplasmic components in the vacuole occur in many plants, suggesting JA–NAC-mediated local cell-fate determination by autophagy may be conserved in angiosperms.

Organogenesis is a well-characterized aspect of flower development that involves the specification and commitment decisions following the switch from a vegetative to a reproductive state[1]. Studying flower organogenesis offers insights into plant development as this stage comprises 1) the transition from an indeterminate to a determinate growth system, that is, the formation of floral organs from groups of undifferentiated meristematic cells that have small vacuoles and few cytosolic components and organelles; and 2) the eventual shedding of these organs to achieve full reproductive competence, such as floral-organ abscission giving way to seed or fruit development following fertilization[2]. On the one hand, the transition to determine growth is akin to the standard notion of development, wherein complexity is borne out of a series of progressions from a relatively simple system. Abscission on the other hand can be considered a less-conventional approach to understanding development because the genetically encoded attrition of cells, tissues and organs may appear to run

counterintuitive to the principle of development. However, petal abscission depends on new RNA and protein synthesis[3], is often accompanied by cell-wall collapse and cytoplasmic and vacuolar reduction[4]. These observations suggests that petal abscission is actively controlled at the cellular level rather than being merely the result of a catastrophic error[5]. In fact, the reprogramming of differentiated cells is essential for plant fitness and plays vital roles in organogenesis, including in floral organogenesis.

*Arabidopsis thaliana* flowers are composed of four sepals, four petals, six stamens and two fused carpels arranged in concentric circles or whorls, specifically in this sequence above[6,7]. Genetic studies with homeotic mutants led to the ABC model of flower development describing how floral organs are specified[6,8,9]. The C-class gene *AGAMOUS* (*AG*, encoding a MADS-box transcription factor) is expressed as early as during stage 3 of flower formation and was initially shown to specify stamen and carpel identity. *AG* is expressed until the later stages of flower development and is confined to specific cell types in the stamens and carpels[10]. AG also regulates stamen development by regulating the expression of the jasmonic acid (JA) biosynthesis gene *DEFECTIVE IN ANTHER DEHISCENCE1* (*DAD1*)[11]. *DAD1* encodes chloroplastic lipase A1, which catalyzes the first committed step in JA biosynthesis[10]. Under low JA levels, JA target genes are repressed by JASMONATE-ZIM DOMAIN (JAZ) proteins, preventing their transcription factors, such as the basic-helix-loop-helix (bHLH) member MYC2, from activating them[12]. JA serves as the substrate of SCF^COI1, a Skp1-Cullin 1-F-box (SCF) E3 ubiquitin ligase specific to JA responses[12]. CORONATINE INSENSITIVE1 (COI1) is an F-box protein that confers the specificity of SCF^COI1 by enabling JAZ repressors to be targeted for degradation by ubiquitination[12]. JAZ proteins act with their co-repressors TOPLESS (TPL) to repress JA target genes. The JAZ–TPL repressor complex is thought to act by altering chromatin conformation through its interaction with histone deacetylases (HDACs), which promote chromatin compaction by removing acetyl groups from histones, leading to reduced chromatin accessibility[13]. In the presence of JA, JAZ repressors are ubiquitinated by the SCF^COI1–JA-Ile complex and degraded, releasing MYC2 from TPL-imposed repression[14,15]. MYC2 then kicks off the early JA response by recruiting the Mediator complex for transcriptional activation through the Mediator subunit MED25[14–16]. MED25 interacts with COI1 and provides a direct link for how COI1 modulates RNA polymerase II (Pol II) machinery for activating MYC2 targets in response to JA[15,16]. Recent studies have affirmed the role for JA in floral-organ abscission: JA biosynthesis and signaling are determinants of floral-organ abscission timing[17,18]. However, it remains unknown whether the JA pathway also contributes to petal abscission.

Abscission is a physiological process involving the shedding or separation of plant organs from the main plant body, such as petals, leaves and fruits[2,19–21]. Abscission is regulated by endogenous developmental signals and environmental cues for overall maintenance of plant architecture and fitness, and is considered a specific type of programmed cell death[2,20]. *BIFUNCTIONAL NUCLEASE1* (*BFN1*), a marker of programmed cell death, is expressed in the abscission zone during floral-organ abscission[20]. Among multiple signaling pathways, reactive oxygen species (ROS) is one component that triggers programmed cell death. Although floral-organ abscission and flower senescence are distinct processes, senescence typically precedes abscission. Thus, senescence markers can be used as indicators of subsequent abscission. The NAC (NAM, ATAF1/2, CUC2) family of transcription factors is implicated in petal senescence[22]. *NAC* induction was reported in senescent petals in Arabidopsis[22,23]. However, whether NACs also have roles in petal abscission is not known.

Senescence and cell separation are tightly linked with the cellular self-digestion process of autophagy. Autophagy is a degradation system conserved in eukaryotes that removes unwanted cellular components and maintains cellular homeostasis. In plants, up-regulation of autophagy genes occurs in senescent petals of petunia (*Petunia*

*hybrida*) and Japanese morning glory (*Ipomoea nil*)[22,24]. Upon activation of autophagy, a double-membrane structure, the autophagosome, forms around the cargo to be degraded. Autophagosomes relocate from the cytoplasm to the vacuole for cargo degradation. One of the key sets of proteins, which participate both in autophagosome formation and cytoplasm-to-vacuole transport, is the autophagy-related (ATG) proteins. ATG proteins maintain a basal level of autophagy activity under normal conditions but can be induced in response to developmental and environmental cues. A hallmark of Arabidopsis autophagy-deficient mutants is early senescence[25,26]. Live-imaging studies of *atg* mutants revealed a role for ATG proteins in root-cap detachment in response to developmental cues[27]. During petal senescence in Japanese morning glory and carnation (*Dianthus caryophyllus*), numerous vesicles and cytoplasmic components localize to the vacuoles, suggestive of autophagy[4,28]. However, how autophagy in petals is triggered and whether autophagy is linked to petal senescence have not been addressed. Here, we demonstrate how hormone-mediated epigenetic regulation triggers terminal cell differentiation with precisely defined timing, in a specific region of plant tissue, to promote petal abscission in Arabidopsis.

## Results

### AGAMOUS and JA promote petal abscission through cell differentiation at the petal base

Toward understanding the role of AG and its targets specifically during petal abscission, we quantified petal-abscission timing in the flowers of wild-type (WT) Col-0, *ag* and JA-deficient mutant plants (Fig. 1a–j). The youngest flower with visibly emerging floral organs was designated position 0 and was used as reference for assigning flower developmental stages[2] (Fig. 1a and Supplementary Fig. 1). At position –1, WT petals were covered with sepals (Fig. 1b). Rapid petal growth led to normal petal protrusion in WT flowers at position 0 (Fig. 1a, b and Supplementary Fig. 1)[7]. As reported previously, most petals in WT had been shed by position +5 under our growth conditions[29] (Fig. 1a–c and Supplementary Fig. 1). We examined the positions of the flowers when petals detached and observed statistically significant delays in petal abscission for the *ag*, *dad1*, *aos* (lacking ALLENE OXIDE SYNTHASE function) and *coi1* mutants (Fig. 1c and Supplementary Fig. 1). In contrast to WT, *ag* and JA-deficient mutants retained their petals up to position 7 or even later (Fig. 1a–c and Supplementary Fig. 1). In JA-deficient mutants, petal and stamen abscission were delayed (Supplementary Fig. 2).

To estimate the extent of cellular damage in petal cells, we performed 3,3′-diaminobenzidine (DAB) staining for hydrogen peroxide and trypan blue staining for dead cells in petals. Hydrogen peroxide accumulated locally at the bases of petals in all genotypes, but with varying staining intensities (Supplementary Fig. 3). We used petal bases from flowers at position +3, which corresponds to the time immediately before petal abscission in the WT. DAB staining was weaker in petal bases of the *ag* and JA-deficient mutants compared to the WT (Fig. 1d–f). To determine whether this decrease was due to a delay in the onset of DAB accumulation or to defects in ROS production in JA-deficient mutants, we conducted a developmental time course of DAB staining in WT and *dad1*. DAB staining patterns were similar between the WT and *dad1* until position 0 (Supplementary Fig. 4). From position +2 to position +4, DAB staining was stronger in WT petal bases compared to *dad1*[30] (Supplementary Fig. 4). WT petals appeared to shed after reaching a peak in DAB signal (Fig. 1a–c and Supplementary Fig. 4). By contrast, *dad1* petal bases continued to show strong DAB staining from position +4 to position +6, with a plateau at position +8 that largely correlated with the timing of petal abscission in this mutant (Supplementary Fig. 4). WT, *ag* and JA-deficient mutants all have comparable levels of DAB staining before petal abscission (Supplementary Fig. 5). These observations suggest that the weaker DAB staining in *ag* and JA-deficient petal bases from

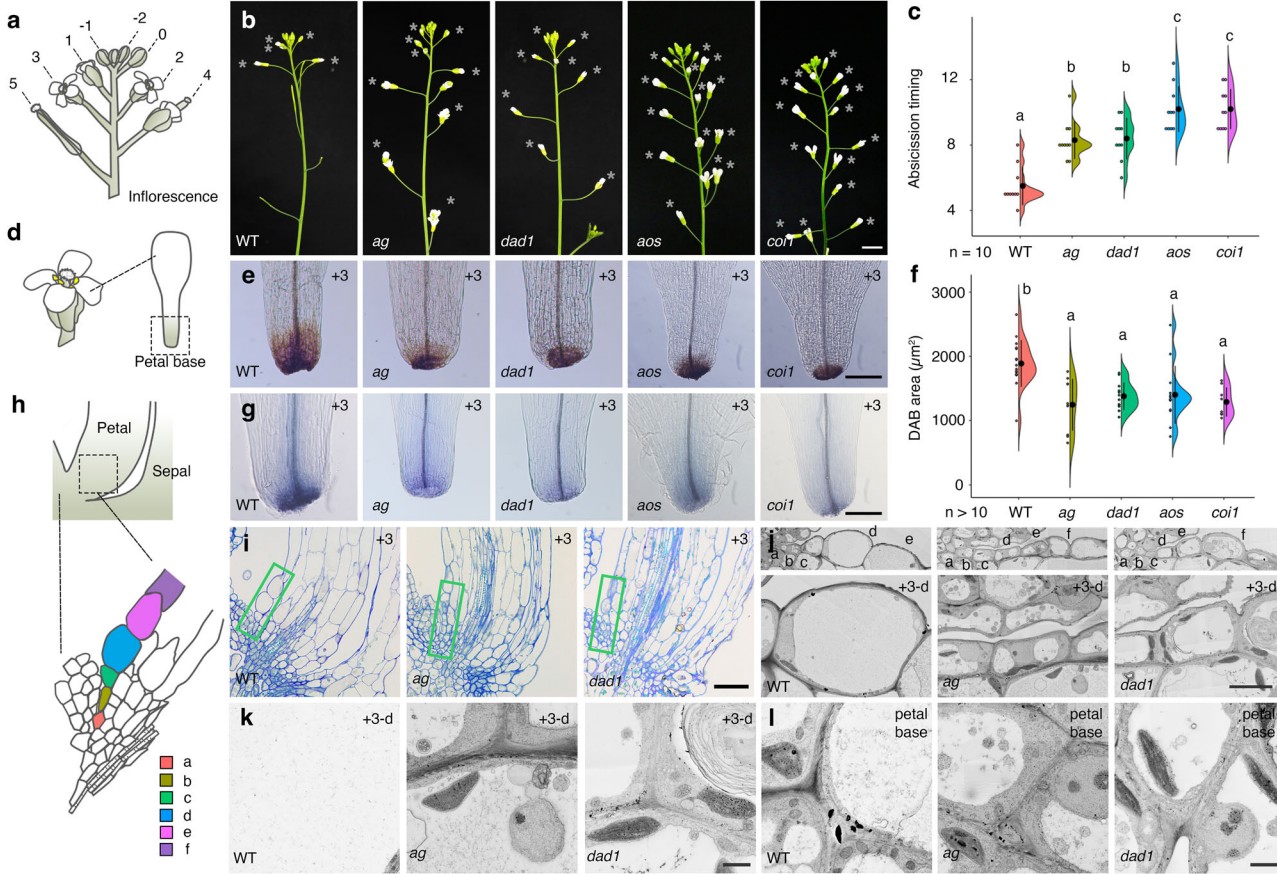

**Fig. 1 | AG-mediated jasmonic acid homeostasis controls the timing of petal abscission. a** Definition of petal-abscission stages. The youngest flower with visible organs is defined as position 0. **b** Side view of wild-type (WT, Col-0), *ag*, *dad1*, *aos* and *coi1* inflorescences. Opened flowers with petals are indicated by gray asterisks. Scale bar = 1 cm. **c** Quantification of abscission timing, shown as individual data points (left) and violin plots (right) for each genotype. Black dots and vertical lines indicate mean and standard deviation (SD), respectively. *n* = 10. Different letters indicate statistically significant differences based on one-way ANOVA test (*p* = 1.3 ×10^{-10}) and post-hoc Tukey's HSD test (*p* < 0.05: See the Data Source file for all combinations of the exact *p*-values). **d** Diagram of the petal base from a position +3 flower. **e** DAB staining of WT, *ag*, *dad1*, *aos* and *coi1* petals for position +3 flowers. Scale bar = 100 μm. **f** Quantification of DAB-stained area (μm²), shown as individual

data points (left) and violin plots (right) for each genotype. Black dots and vertical lines indicate mean and SD, respectively (WT: *n* = 16; *ag*: *n* = 10; *dad1*: *n* = 12; *aos*: *n* = 13; *coi1*: *n* = 8). Different letters indicate statistically significant differences based on one-way ANOVA test (*p* = 7.8 ×10^{-5}) and post-hoc Tukey's HSD test (*p* < 0.05: See the Data Source file for all combinations of the exact *p*-values). **g** Trypan blue staining of wild-type, *ag*, *dad1*, *aos* and *coi1* petals at position +3 flowers. Scale bar = 100 μm. **h** Definition of cellular position during petal abscission. **i** Longitudinal sections of petal base in the wild type, *ag* and *dad1* at position +3. Scale bar = 40 μm. **j** SEM images of petal base in the wild type, *ag* and *dad1* at position 3. Top, Higher magnification of green square in **i**. Bottom, Higher magnification of position d cells. Scale bar = 5 μm. SEM images of position d cells (**k**) and neighboring cells (**l**) at position +3 petals in the wild type, *ag* and *dad1*. Scale bars = 1 μm.

flowers beginning at position +3 could be attributed to a delay in DAB accumulation. We next visualized dead cells with trypan blue staining using position +3 flowers. WT petals showed staining at the petal base and in xylem cells near the petal base, as described previously[31] (Fig. 1g and Supplementary Fig. 3). Consistent with DAB staining, we detected weak trypan blue staining in *ag* and JA-deficient mutants at the same petal stage (Fig. 1g).

To gain more insight into petal abscission, we sectioned the petal base from position +3 flowers and designated the first epidermal cell located at the base on the adaxial side as position 'a' (Fig. 1h). At position a, WT, *ag* and *dad1* cells were of comparable sizes (Fig. 1h–j), contained electron-dense material, and were primarily filled with large nuclei and cytoplasm (Fig. 1j; top). At position d, WT petal cells were partially filled with large vacuoles, indicating cell differentiation (Fig. 1j; top). Notably, we observed partial vacuolation of *ag* and *dad1* cells at position d from flowers at position +3 (Fig. 1h–j). Meanwhile, *dad1* cells at position d from flowers at position +7 had vacuolation similar to that of WT cells at the same position before petal abscission (Supplementary Fig. 6). Furthermore, round membrane structures containing cytoplasmic components were observed within vacuoles in the *ag* or

JA-related mutant backgrounds (Fig. 1k,l). This observation suggests that vacuolar-trafficking pathways are affected in these mutants and that the extent of vacuolation is correlated with the timing of petal abscission. In summary, AGAMOUS and jasmonic acid promote petal abscission through cellular differentiation at petal bases.

## JA accumulates at the petal base and triggers cell differentiation

To explore the effects of JA on the timing of petal abscission and petal-cell differentiation, we characterized the effects of exogenous JA application (Fig. 2). Because *ag* and *dad1* mutants accumulate less JA[11,30], we examined the extent of phenotypic rescue on these mutants following a single spray application of 500 μM methyl jasmonate. Consistent with a previous report[30], this regimen rescued the petal-abscission defect without causing visible secondary effects. In the WT, the timing of petal abscission was similar regardless of JA treatment (Fig. 2a, b). JA treatment also had no obvious effect on DAB or trypan blue staining in WT petals at position +3, suggesting that the WT has sufficient JA to trigger petal abscission (Fig. 2c–e). By contrast, JA treatment rescued the delayed petal-abscission phenotype in the *ag* and *dad1* mutants (Fig. 2a, b), restored DAB and trypan blue staining,

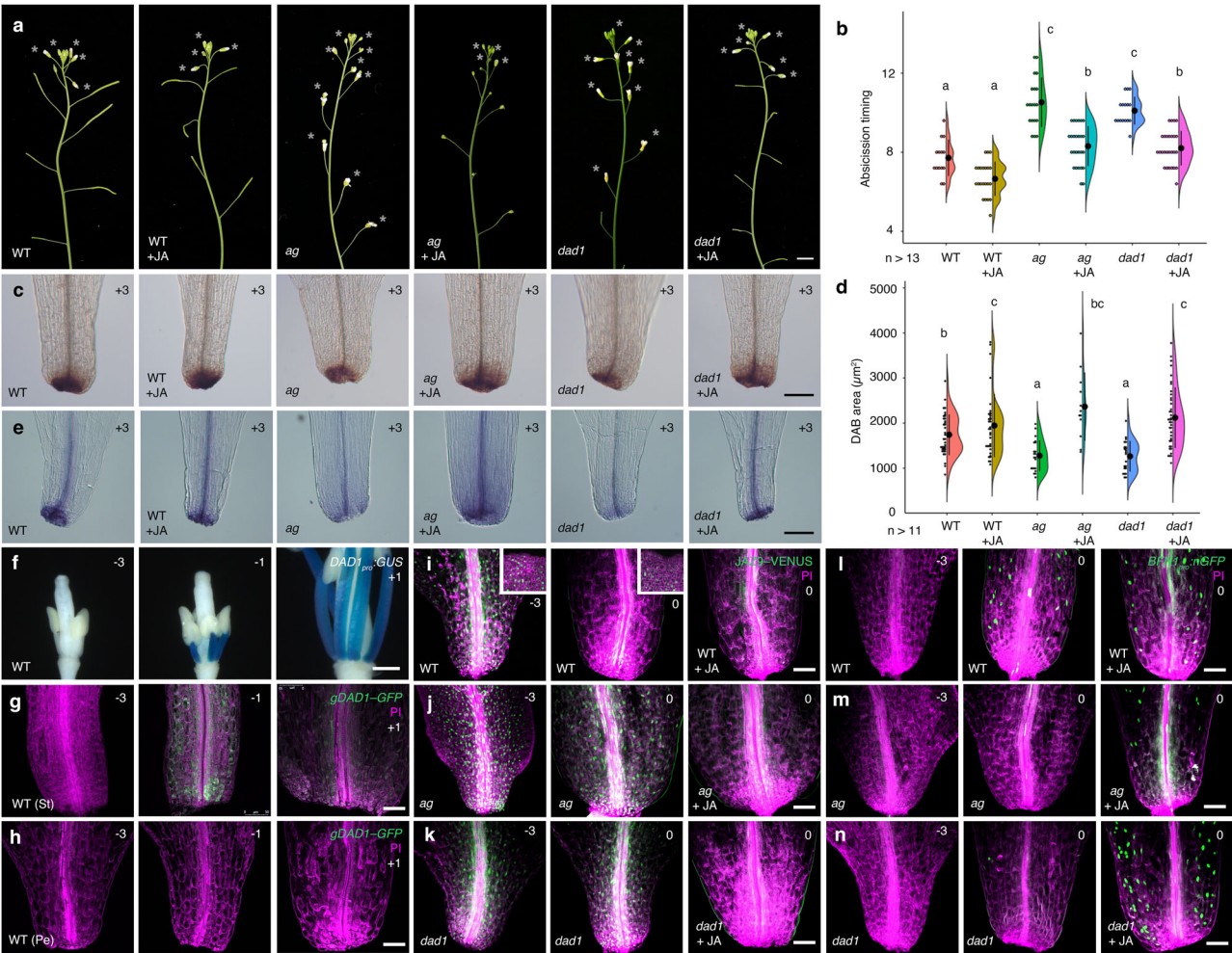

**Fig. 2 | Proper regulation of jasmonic acid responses and programmed cell death at the petal base control the timing of petal abscission. a** Profile view of mock- and jasmonic acid (JA)-treated WT, *ag* and *dad1* inflorescences. Opened flowers with petals are indicated by gray asterisks. Scale bar = 1 cm. **b** Quantification of abscission timing, shown as individual data points (left) and violin plots (right) for each genotype (WT: $n = 14$; WT + JA: $n = 22$; *ag*: $n = 19$; *ag* + JA: $n = 26$; *dad1*: $n = 16$; *dad1* + JA: $n = 27$). Different letters indicate statistically significant differences based on one-way ANOVA test ($p = 1.1 \times 10^{-16}$) and post-hoc Tukey's HSD test ($p < 0.05$: See the Data Source file for all combinations of the exact *p*-values). **c** DAB staining of mock- and JA-treated WT, *ag* and *dad1* petals at position +3 flowers. Scale bar = 100 μm. **d** Quantification of DAB-stained area (μm²), shown as individual data points (left) and violin plots (right) for each genotype. Black dots and vertical lines indicate mean and SD, respectively (WT: $n = 36$; WT + JA: $n = 34$; *ag*: $n = 21$; *ag* + JA:

$n = 12$; *dad1*: $n = 22$; *dad1* + JA: $n = 40$). Different letters indicate statistically significant differences based on one-way ANOVA test ($p = 2.3 \times 10^{-10}$) and post-hoc Tukey's HSD test ($p < 0.05$: See the Data Source file for all combinations of the exact *p*-values). **e** Trypan blue staining of mock- and JA-treated WT, *ag* and *dad1* petals from position +3 flowers. Scale bar = 100 μm. **f** *DAD1pro:GUS* staining pattern in WT petals from position −3 (left), −1 (middle), and +1 (right) flowers. Scale bar = 500 μm. gDAD1−GFP accumulation in WT stamens (**g**) and petals (**h**) at position −3 (left), −1 (middle), and +1 (right) flowers. Magenta: cell wall; green: DAD1−GFP. Scale bar = 50 μm. **i−k** JAZ9−VENUS accumulation in mock- (left and middle) and JA-treated (right) WT, *ag*, and *dad1* petals from position −3 (left) and 0 (middle and right) flowers. **l−n** *BFN1pro:nGFP* expression in mock- (left and middle) and JA-treated (right) WT, *ag* and *dad1* petals from position −3 (left) and 0 (middle and right) flowers. Magenta: cell wall; green: JAZ9−VENUS or GFP. Scale bars = 50 μm.

and cellular vacuolation to WT levels in mutant petals (Fig. 2c−e and Supplementary Fig. 7). We conclude that delayed petal abscission in *ag* and *dad1* may be at least in part due to lower JA levels.

Next, we examined the spatiotemporal pattern of JA biosynthesis and accumulation in floral organs using a 3.5-kb *DAD1* promoter fragment driving *ß-glucuronidase* (*GUS*) expression[30]. We did not detect GUS staining from the *DAD1pro:GUS* reporter in stamen filaments at position −3 but observed its gradual activation from position −2 onward (Fig. 2f and Supplementary Fig. 8). The strongest GUS staining was seen in petal bases at position +1 and +2, but GUS was not detected in petals (Supplementary Fig. 8). Similar to *DAD1*, other JA biosynthesis genes, such as *AOS* and *OPR3*, were also expressed primarily in stamen filaments based on their respective GUS reporter activities (Supplementary Fig. 9)[32,33]. We next cloned the *DAD1* genomic region in-frame with *GFP* (green fluorescent protein) and introduced the resulting construct into the *dad1* mutant. This transgene fully restored petal

abscission, and DAB and trypan blue staining to WT levels, indicating the DAD1−GFP fusion protein is functional (Supplementary Fig. 10). *DAD1* promoter activity (*DAD1pro:GUS*) and DAD1−GFP fluorescence overlapped during the same stages of flower development (Fig. 2f, g). Furthermore, DAD1−GFP fluorescence was not seen in petals (Fig. 2h).

We then analyzed fluorescence from the JA-perception biosensor JAZ9−VENUS[34]. In the WT, JAZ9−VENUS accumulated to high levels at the base of petals from position −3 flowers (Fig. 2i). As DAD1−GFP started to accumulate in stamen filaments of position −1 flowers, JAZ9−VENUS diminished in petals by position 0, which we interpret as a response to JA accumulation (Fig. 2i). As in the WT, we observed high JAZ9−VENUS fluorescence in *ag* and *dad1* at petal bases of position −3 flowers (Fig. 2j, k). However, JAZ9−VENUS remained high in *ag* and *dad1* in position 0 flowers, suggesting that AG and DAD1 are required for JAZ9 degradation at the petal base. Treating WT petals with JA should promote degradation of the JAZ9−VENUS sensor. Indeed, we

detected little JAZ9–GFP fluorescence in the petal base of WT flowers from position 0 (Fig. 2i), as well as in *ag* and *dad1* petals at position 0 (Fig. 2j, k). These results indicate that the delay of petal abscission in *ag* and *dad1* is caused by decreased JA levels in petal bases.

We also examined the link between JA accumulation and subsequent cell differentiation. Because programmed cell death is often involved in organ abscission, we used the programmed-cell-death-associated gene *BFN1* as a marker[35–37]. In WT petal bases, GFP fluorescence was not detected from the *BFN1pro:nGFP* transgene in position −3 flowers (Fig. 2l). Consistent with the timing of JA accumulation, *BFN1* began to be expressed in WT petal bases at position 0 (Fig. 2l). In *ag* and *dad1* petals, GFP fluorescence was either absent (position −3) or very faint (position 0) (Fig. 2m, n), but JA treatment restored strong GFP fluorescence in the petal bases of flowers from position 0 of *ag* and *dad1* mutants (Fig. 2m, n). Our results thus suggest that AG- and DAD1-mediated JA accumulation triggers cell differentiation at the petal base for proper petal abscission.

### JA signaling mediates expression of key JA homeostasis targets during petal abscission

To identify JA-regulated genes required for petal abscission, we performed RNA-seq of WT and *dad1* plants, before comparing this dataset to publicly available datasets specific to JA signaling and responses (Supplementary Data 1). We identified 779 statistically significantly down-regulated genes in *dad1* relative to the WT (Fig. 3a and Supplementary Data 2). We confirmed the down-regulation of selected genes in *dad1* by reverse-transcription quantitative PCR (RT–qPCR) (Supplementary Fig. 11). We then compared 779 genes to three datasets: one from a decuple *jaz* mutant (*jazD* compared to the WT)[38] and two from a time course of JA-treated WT plants (for 1 h and 24 h after JA treatment)[16] (Fig. 3a). We saw that 244 of the 779 down-regulated genes in *dad1* are up-regulated in the *jazD* mutant, suggesting that these genes are repressed in the presence of functional JAZ repressors[39,40] (Fig. 3a). Of these 244 genes, 70 were statistically significantly up-regulated upon JA treatment (Fig. 3a and Supplementary Data 3), thus defining a high-confidence set of genes downstream of DAD1 (Fig. 3b).

To understand the potential functions of these 70 genes, we performed a Gene Ontology (GO)-term enrichment analysis[41,42] (Fig. 3c and Supplementary Data 4). Enriched terms included 'sugar transport', 'plant hormone', 'cell death', 'defense response' and 'stress response', consistent with known functions associated with JA (Fig. 3c). One of the 70 genes is *SUGAR TRANSPORT PROTEIN13* (*STP13*), encoding a hexose-specific H⁺-symporter, the expression of which is induced as senescence and programmed cell death progress[43,44]. Additionally, the plant-specific transcription factor ANAC102 contributes to stress responses (with other *ANAC* genes) and was among our high-confidence target list[45]. Refinement of the GO results using REVIGO revealed a classification into three main groups: 'response', 'metabolism' and 'transport' (Fig. 3d). The 'response' category included GO terms related to stress response (Fig. 3d). The 'metabolism' category comprised GO terms related to hormone, amino acid, organic acid, lipid, and toxin metabolism (Fig. 3d). The 'transport' category included 'transport', 'cellular macromolecule localization', 'organic-acid transport', 'carboxylic-acid transport' and 'amine transport' (Fig. 3d).

We then scanned for *cis*-motifs present within 2-kb promoter regions of all 70 genes (Fig. 3e and Supplementary Data 5). G-box (MYC) and E-box (BES) motifs are enriched (Fig. 3e). JA-dependent MYC transcription factors and brassinosteroid-dependent transcription factors such as BRI1-EMS-SUPPRESSOR (BES) and BRASSINAZOLE-RESISTANT (BZR) bind to G-boxes and E-boxes, respectively[14,46–51]. We also identified the two types of abscisic acid (ABA)-responsive element motifs recognized by ABRE BINDING FACTORs (ABFs) and ABA INSENSITIVE (ABI) transcription factors[52–55] (Fig. 3e). These enriched *cis*-elements all contain CACGTG at their core, suggesting that the 70 high-confidence JA-regulated genes may

be regulated by JA-dependent and/or other hormone-dependent transcription factors.

### JA triggers a chromatin-state switch to promote petal abscission

MYC2 is a master regulator of JA-dependent transcription[14,15] and acts cooperatively with MED25 to regulate these networks[56]. We observed delayed petal abscission in the *myc2 myc3 myc4* and *med25* mutants, along with partial vacuolation and weaker DAB and trypan blue staining, compared to the WT (Supplementary Figs. 6, 12, 13). These phenotypes suggest that MYC2 and MED25 may serve as activators of petal abscission. The delay in the appearance of DAB-staining signal in *myc2 myc3 myc4* correlated with the timing of petal abscission (Supplementary Fig. 14). We re-analyzed published chromatin immunoprecipitation (ChIP) data for MYC2 and MED25 against all 70 high-confidence DAD1 downstream genes[57–60] (Fig. 4a). Of these, 66 (94%) were direct MYC2 targets (Fig. 4a), of which 25 genes were shared targets of MYC2 and MED25 (Fig. 4a). Because the MYC2 and MED25 complex activates transcription via epigenetic regulation, including chromatin accessibility[16,56,57,61], we checked for the presence of DNase I-hypersensitive (DNase I HS) sites, and all but one of these 25 genes had chromatin regions hypersensitive to DNase I (Fig. 4a). Ten genes shared equal counts of DNase I-HS sites that are leaf- and flower-specific, while six genes had more leaf-specific sites, and the remaining eight genes had more flower-specific sites (Fig. 4a).

The TRAVA (transcriptome variation analysis) database contains RNA-seq datasets from position −15 to position +2 flowers[62,63], and reflects a gradual up-regulation of most of the 25 genes analyzed across the time course (Fig. 4b). For example, *ETHYLENE RESPONSE FACTOR 11* (*ERF11*), *GLUTAMATE DECARBOXYLASE4* (*GAD4*) and *STP13* expression levels were very low at earlier stages but reached their peaks by position +2 flowers. Several genes, such as *RELATED TO ABI3/ VP1 1* (*RAV1*), *ANAC102*, and *SENESCENCE-ASSOCIATED GENE14* (*SAG14*) also reached their highest expression in position +2 flowers (Fig. 4b). Furthermore, these genes were down-regulated in both JA biosynthetic and signaling mutants (Supplementary Fig. 15). This result suggests that these JA-regulated genes may be involved in petal abscission. Indeed, we detected similar binding peaks for MYC2 and MED25 at the *ANAC102*, *RAV1*, *STP13* and *SAG14* loci from public ChIP-seq datasets[57–60]. Using floral tissues, we confirmed that MYC2 and MED25 bind to these four loci by ChIP–qPCR (Fig. 4c). Based on micrococcal-nuclease digestion followed by sequencing (MNase-seq) data, we mapped nucleosomes mainly to the gene bodies of these four loci (Fig. 4b). Consistent with the physical interaction between MED25 and HISTONE ACETYLTRANSFERASE1 (HAC1), we also observed H3K9ac enrichment in the four gene bodies (Fig. 4b). In addition, we detected DNase I-HS sites in the *RAV1* and *SAG14* promoters in flowers, and in both the promoters and gene bodies of *ANAC102* and *STP13* (Fig. 4b). These binding patterns suggest that a JA-mediated chromatin-state switch plays a key role in target-gene activation.

To test this hypothesis, we used the JA-deficient mutant *dad1* to probe chromatin state when JA signaling is inactive. Previous studies suggested that MYC2-interacting JAZ proteins recruit the transcriptional co-repressor TPL and the histone deacetylase HDA19 to MYC2 target loci to prevent MYC2 from activating its target genes when JA levels are low[14]. We immunoprecipitated more TPL and HDA19 from the *ANAC102* and *STP13* loci in floral tissues in the absence of JA (*dad1* mutant) than in the presence of JA (WT) (Fig. 4f, g and Supplementary Fig. 16). Notably, another histone deacetylase HDA6 did not have detectable binding at either *ANAC102* or *STP13* loci, suggesting that specific combinations of HDA proteins may be important for petal abscission[64] (Supplementary Fig. 17). To test whether MYC2 recruits MED25 to its target loci to overcome the repressed chromatin state, we performed ChIP with an anti-GFP antibody on WT and *dad1* flowers harboring the *MED25–GFP* transgene. We detected less chromatin from the

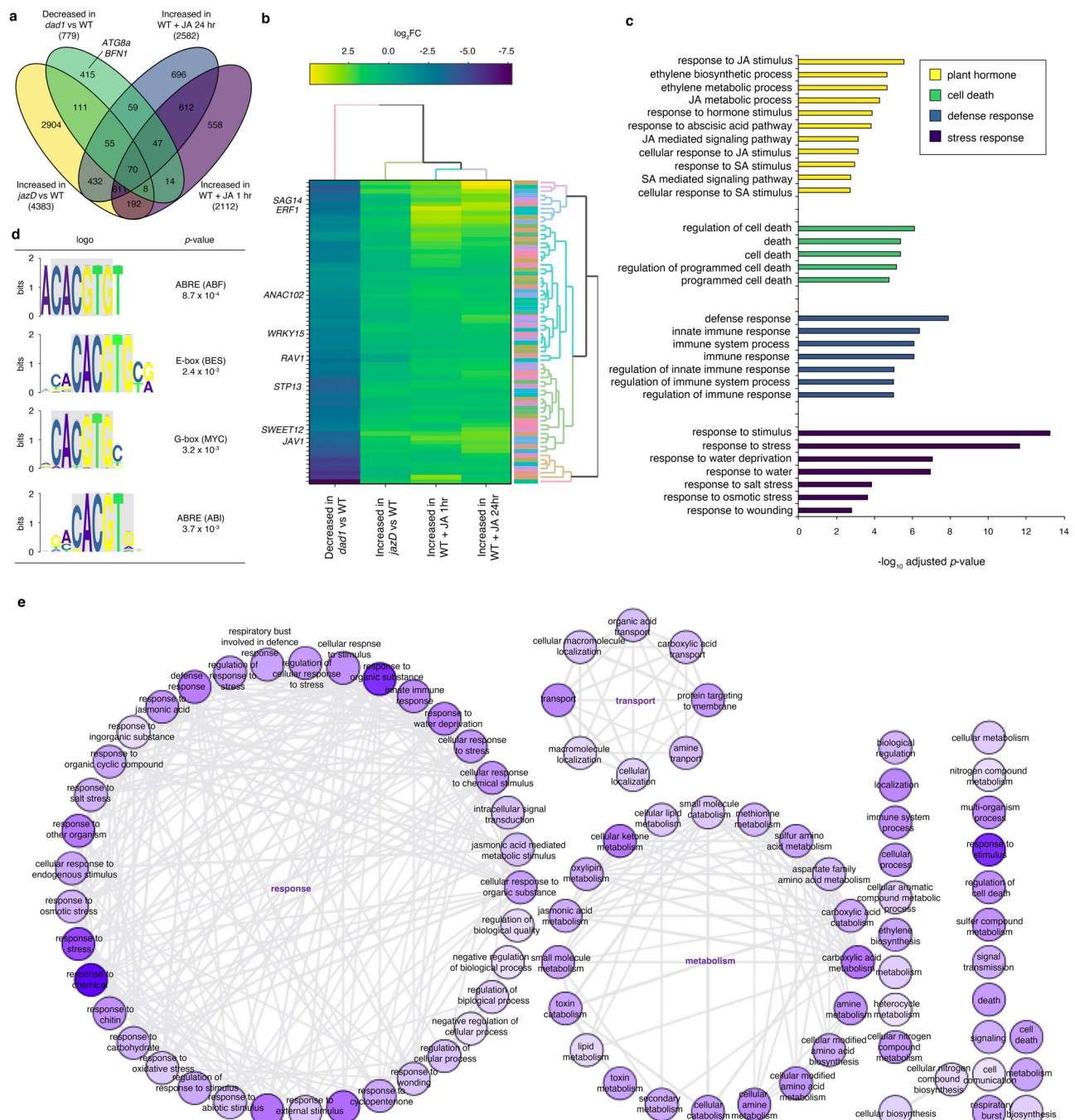

**Fig. 3 | Jasmonic acid-mediated gene expression during petal abscission. a** Venn diagrams showing the overlap between genes down-regulated in *dad1* vs. the WT, up-regulated in *jazD* vs. the WT, up-regulated in the WT after 1 h JA treatment, and up-regulated in the WT after 24 h JA treatment. **b** Hierarchical clustering of 70 high-confidence JA-regulated genes during petal abscission. The heatmap shows the Log₂ fold-change of these genes in the RNA-seq data in **a. c** GO-term enrichment analysis using the above 70 high-confidence JA-regulated genes. Selected GO terms related to plant hormones, cell death, defense response and stress response are shown with their associated −Log₁₀ adjusted *p*-value. The *p*-values for GO analysis are obtained by converting the Z-scores based on a two-tailed Z-test. **d** *Cis*-motif enrichment in 70 high-confidence JA-regulated genes during petal abscission. Sequence-logo representation of nucleic acid multiple sequence alignment and −Log₁₀ adjusted *p*-values converted from Z scores based on a two-tailed Z-test are shown. **e** Interactive graph view of GO terms generated by REVIGO. The node radius indicates the specificity, and the color shading corresponds to the *p*-values. Highly similar GO terms are linked by edges, with edge thickness indicating the degree of similarity.

*ANAC102* and *STP13* loci associated with MED25–GFP when performing ChIP on *dad1* flowers relative to the WT (Fig. 4h and Supplementary Fig. 16). Furthermore, the WT had greater H3K9ac and RNA Pol II enrichment at the *ANAC102* and *STP13* loci than in *dad1*, possibly due to functional JA-signaling activity (Fig. 4i, j and Supplementary Fig. 16). To examine chromatin accessibility, we assayed open and nucleosome-depleted regions by formaldehyde-assisted isolation of regulatory elements (FAIRE). Chromatin accessibility of the MYC2-bound regions at the *ANAC102* and *STP13* loci in the WT was higher than in *dad1* (Fig. 4k), suggesting that MYC2 recruits MED25 to increase accessibility of the DNA and induce the expression of abscission regulators.

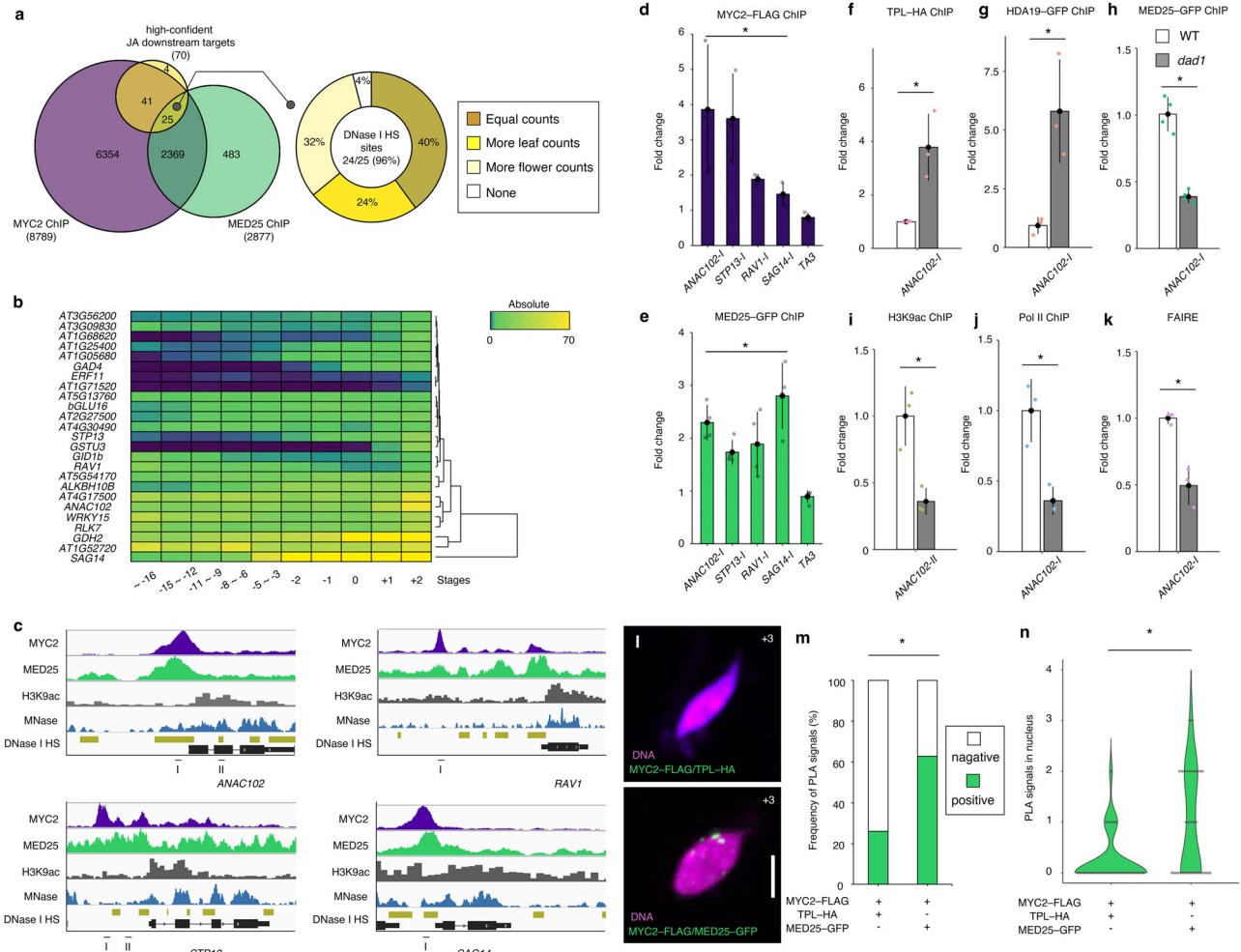

**Fig. 4 | A jasmonic acid-mediated chromatin-state switch controls petal abscission. a** Left, Venn diagram showing the overlap between direct MYC2 targets, direct MED25 targets, and 70 high-confidence JA-regulated genes. Twenty-five out of 70 high-confidence JA-regulated genes are common direct targets of MYC2 and MED25. Right, pie chart of DNase I-hypersensitivity assay for 25 high-confidence JA-regulated genes. **b** Heatmap representation of the absolute expression values of the 25 above genes based on public transcriptome data. **c**, IGV browser view of MYC2, MED25 and H3K9Ac ChIP-seq signals, MNase-seq signals and DNase I-hypersensitive sites (HSs). Binding of MYC2 (**d**) and MED25 (**e**) at shared loci based on ChIP–qPCR. Positions of PCR amplicons for ChIP-qPCR are shown in **c**. Values are means ± standard error (*n* = 4 independent experiments). Asterisks indicate statistically significant differences between the direct target and the negative control (*TA3*) based on one-tailed Student's *t* test (MYC2 ChIP, *STP13*: *p* = 0.03; *ANAC102*: *p* = 0.03; *RAV1*: *p* = 3.9 ×10⁻⁴; *SAG14*: *p* = 0.03) (MED25 ChIP, *STP13*: *p* = 7.0 ×10⁻⁴; *ANAC102*: *p* = 1.5 ×10⁻³; *RAV1*: *p* = 0.03; *SAG14*: *p* = 4.8 ×10⁻³). Association of TPL–HA (**f**), HDA19–GFP (**g**), MED25–GFP (**h**), H3K9ac (**i**) and Pol II (**j**) at the *ANAC102* promoter in WT and the *dad1* mutant

based on ChIP-qPCR. Positions of PCR amplicons for ChIP-qPCR are shown in **c**. Values are means ± SE (*n* = 3 independent experiments). Asterisks indicate statistically significant differences between WT and *dad1* based on one-tailed Student's *t* test (**f**: *p* = 0.03; **g**: *p* = 0.03; **h**: *p* = 4.2 ×10⁻⁴; **i**: *p* = 0.01; **j**: *p* = 7.4 ×10⁻³). **k** Chromatin accessibility at the *ANAC102* promoter in the WT and the *dad1* mutant based on FAIRE-qPCR. Positions of PCR amplicons for ChIP-qPCR are shown in **c**. Values are means ± SE (*n* = 3 independent experiments). Asterisks indicate statistically significant differences between the wild type and *dad1* based on one-tailed Student's *t* test (*p* = 0.01). **l** Protein–protein interaction between MYC2 and the repressor TPL or the activator MED25 in the nuclei from WT petals at position 3. Magenta: DNA; green: PLA signals detected as dots. Scale bar = 5 μm. **m** Frequency of petal cell nuclei with PLA signals. *p*-values were determined by one-sided Chi-squared test. *p* < 0.05. **n** Quantification of number of PLA signals, shown as violin plots. *n* = 33. Asterisks indicate statistically significant differences between MYC2–TPL and MYC2–MED25 interaction based on two-tailed Student's *t* test.

MYC2 forms a TPL co-repressor/histone deacetylase complex and an activator complex with MED25[65]. To visualize protein interactions in petals at specific stages, we employed an in situ proximity-ligation assay (PLA) using petals from position +3 flowers and quantified interaction foci per nucleus[66]. Over 70% of nuclei had no detectable PLA signal probed with anti-GFP and anti-HA antibodies in *MYC2–FLAG TPL–HA* transgenic petals. By contrast, we detected at least one focus per nucleus by in situ PLA in *MYC2–FLAG MED25–GFP* transgenic petals, suggesting that MYC2 tends to associate more with the MED25 activator complex than with the TPL co-repressor/histone deacetylase complex in petals at position +3 (Fig. 4k–n).

## *ANAC102* is a shared target of MYC2 and MED25 and controls petal abscission

Of the 25 shared target genes between MYC2 and MED25, we focused on *ANAC102* and first determined the extent of conservation in related genes with Gene Slider and phylogenetic shadowing[67–71]. The DNA region bound by MYC2 and MED25 at the *ANAC102* locus was highly conserved across nine *Brassica* species (Fig. 5a and Supplementary Fig. 18) and contains three conserved G-box motifs (Fig. 5b). To test the regulatory relevance of these motifs, we placed *GUS* under the control of the *ANAC102* promoter (*ANAC102pro:GUS*) or a version lacking all three G-boxes (*ANAC102Δpro:GUS*). *ANAC102* was expressed in petal-base cells and vasculature cells immediately prior to petal abscission at

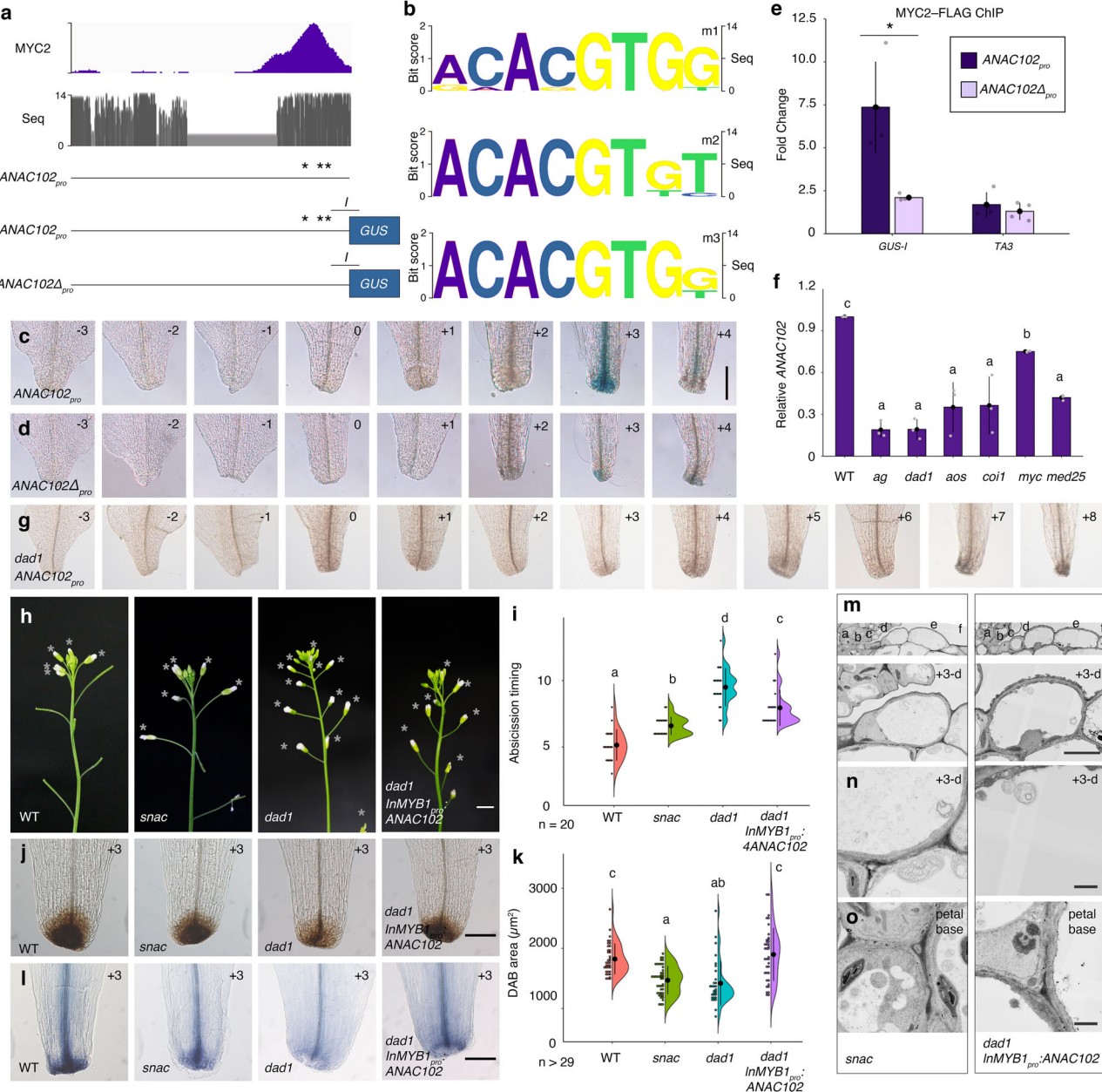

**Fig. 5 | ANAC102 controls petal abscission. a** Identification of evolutionarily conserved G-box motifs at the *ANAC102* promoter. Top, IGV browser view of MYC2 binding peaks by ChIP-seq. Middle, Bit score comparison of 5′ regions of the promoter between nine *Brassicaceae* species by Gene Slider. Bottom, diagram of the *ANAC102* promoter with and without mutations in the G-box motifs (shown as asterisks). PCR amplicons for ChIP–qPCR are shown in **e. b** Web logos of three evolutionarily conserved G-box motifs. Reporter gene expression of intact (**c**) and G-box mutants of the *ANAC102* (*ANAC102Δ*) (**d**) promoter. Scale bar = 100 μm. **e** Association of MYC2 with the *ANAC102pro:GUS* and *ANAC102Δpro:GUS* transgenes based on ChIP–qPCR. Values are means ± SE (*n* = 4 independent experiments). Asterisks indicate statistically significant differences of MYC2 binding between the wild-type and mutated (Δ) *ANAC102* promoter based on two-tailed Student's *t* test. **f** Relative *ANAC102* expression in WT and JA-related mutants, as determined by RT–qPCR. Values are means ± SE (*n* = 3 independent experiments). Different letters indicate statistically significant differences based on one-way ANOVA ($p = 1.9 \times 10^{-6}$) and post-hoc Tukey's HSD test ($p < 0.05$: See the Data Source file for all combinations of the exact *p*-values). **g** GUS reporter gene expression of the *ANAC102* promoter in the *dad1* mutant. Scale bars = 100 μm. **h** Side view of wild-type, *snac* septuple, *dad1* and *dad1 InMYB1pro:ANAC102* inflorescences. Opened flowers with petals are indicated by asterisks. Scale bar = 1 cm. **i** Quantification of abscission timing, shown as individual data points (left) and violin plots (right) for each genotype. Black dots and vertical lines indicate mean and SD, respectively. *n* = 20. Different letters indicate statistically significant differences based on one-way ANOVA test ($p = 1.1 \times 10^{-16}$) and post-hoc Tukey's HSD test ($p < 0.05$: See the Data Source file for all combinations of the exact *p*-values). **j** DAB staining of WT, *snac*, *dad1* and *dad1 InMYB1pro:ANAC102* petals from position +3 flowers. Scale bar = 100 μm. **k** Quantification of DAB-stained area (μm²), shown as individual data points (left) and violin plots (right) for each genotype. Black dots and vertical lines indicate mean and SD, respectively (WT: *n* = 37; *snac* septuple: *n* = 45; *dad1*: *n* = 30; *dad1 InMYB1pro:ANAC102*: *n* = 38). Different letters indicate statistically significant differences based on one-way ANOVA test ($p = 9.9 \times 10^{-11}$) and post-hoc Tukey's HSD test ($p < 0.05$: See the Data Source file for all combinations of the exact *p*-values). **l** Trypan blue staining of WT, *snac*, *dad1* and *dad1 InMYB1pro:ANAC102* petals of position +3 flowers. Scale bar = 100 μm. **m** SEM images of petal base in *snac* and *dad1 InMYB1pro:ANAC102* plants from position +3 flowers. Top, Higher magnification of the green square in Fig. 1i. Bottom, Higher magnification of position d cells. Scale bar = 5 μm. EM images of position d cells (**n**) and neighboring cells (**o**) from position +3 petals in *snac* and *dad1 InMYB1pro:ANAC102* plants. Scale bar = 1 μm.

positions +3 and +4 (Fig. 5c and Supplementary Fig. 19). This transient *GUS* induction was much lower from the *ANAC102Δpro* construct (Fig. 5c and Supplementary Fig. 20), with only a faint signal in petals at positions +3 and +4 (Fig. 5c). We next exploited an in vivo method to test MYC2 binding to intact or mutated promoters[72]. By designing promoter- and transgene-specific primers, we detected strong MYC2–FLAG binding at the *ANAC102pro* transgene, but much weaker binding to *ANAC102Δpro*, suggesting that MYC2 uses conserved G-box motifs to directly bind to the *ANAC102* promoter and activate its expression at petal bases immediately prior to petal abscission.

In petals at position +3, *ANAC102* was expressed at lower levels in mutants lacking direct upstream ANAC102 regulators and in JA biosynthesis- and perception-deficient mutants compared to the WT (Fig. 5f). We crossed the *ANAC102pro:GUS* reporter into the *dad1* mutant and did not detect GUS signal in *dad1* petals at position +3, at a time when the reporter-construct signal reaches its peak in the WT (Fig. 5c, g). We also did not observe *ANAC102* induction in *dad1* petals even after position +3. These results suggest that *ANAC102* functions downstream of the JA pathway during petal abscission.

To understand the role of *ANAC102* during petal abscission, we used the previously published stress-responsive *nac* septuple mutant (*snac*)[45]. Because several *NAC* family members were down-regulated in the *dad1* mutant, we hypothesized that not only *ANAC102*, but also other NACs might have roles during petal abscission (Supplementary Fig. 21). Compared to the WT, timing of petal abscission was statistically significantly delayed in the *snac* mutant (Fig. 5h, i and Supplementary Fig. 22). In agreement with ANAC102 being a key ROS-signaling regulator[73], DAB and trypan blue staining was weaker in *snac* petals at position +3 (Fig. 5j–l). As with the JA-deficient mutants, DAB staining in the *snac* mutant was stronger than in WT before petal abscission. This suggests that the *snac* mutant has a delay in ROS accumulation rather than a general decrease in ROS production at petal bases (Supplementary Fig. 23). Petal sections revealed that +3 petals at position d are less vacuolated in the *snac* mutant than in the WT (Figs. 1i, j and 5m, n). Like the *ag* and *dad1* mutants, we detected more vesicles in *snac* septuple mutant vacuoles than the WT (Fig. 5n, o).

To further evaluate the function of ANAC102 downstream of the JA pathway, we attempted to rescue the *dad1* mutant with *ANAC102* placed under the control of a heterologous petal-specific promoter from Japanese morning glory, *InMYB1* (*InMYB1pro:AtANAC102*)[74–77] (Supplementary Fig. 24). Indeed, *ANAC102* was expressed higher in petals at position +3 of *dad1 InMYB1pro:AtANAC102* flowers than in *dad1* mutant flowers, although not as much as in the WT (Supplementary Fig. 25). The *InMYB1pro:AtANAC102* construct partially rescued the delayed timing of petal abscission of *dad1* and the DAB and trypan blue staining pattern at the petal base of position +3 flowers (Fig. 5h–l and Supplementary Figs. 22, 23). Consistent with this result, we observed proper vacuolation in +3 petals of *dad1 InMYB1pro:AtANAC102* flowers at position d, as with the WT (Figs. 1i, j and 5m, n). Furthermore, the accumulation of autophagic bodies in petal bases of the *snac* mutant was rescued to WT levels in a transgenic *dad1 InMYB1pro:AtANAC102* line (Fig. 5n, o). We conclude that ANAC102 functions downstream of the JA pathway to control petal abscission through cell differentiation at its base.

### ANAC102 orchestrates diverse cellular processes during petal abscission

To identify target genes regulated by ANAC102 during petal abscission, we generated *gANAC102−GFP* transgenic plants that carry the entire *ANAC102* genomic region (promoter and coding region) fused to *GFP*. ANAC102−GFP specifically accumulated in cells at the base of petals in position 3 flowers (Fig. 6a and Supplementary Fig. 26). We also detected GFP fluorescence in the vasculature of petal bases, as seen earlier with the *ANAC102pro:GUS* reporter (Figs. 5c and 6a). We

isolated inflorescences containing flowers up to position +3 for anti-GFP immunoprecipitation and ChIP-seq. In total, we identified 4,196 significant ANAC102 ChIP peaks (MACS2 summit *q*-value < $10^{-10}$) (Fig. 6b and Supplementary Data 6). Most ANAC102 peaks were located near the transcription start site, with occasional peaks also around transcription termination sites (Supplementary Fig. 27). De novo motif analysis using ChIP-peak summits identified several *cis*-elements previously ascribed to ANAC transcription factors from DNA affinity-purification sequencing (DAP-seq) analysis[78,79] (Supplementary Data 7). Although in vitro ANAC102 binding was not examined by the previous cistrome, our motif analysis identified a *cis*-element bound directly by ANAC047, which belongs to the same SNAC family (Fig. 6c) (*p*-value < $10^{-22}$). Furthermore, NAM and ANAC005 *cis*-elements were also included (*p*-value < $10^{20}$) (Fig. 6c). These results suggest that ANAC102 targets loci via these *cis*-elements.

To explore the ANAC102-regulated transcriptional program in petal abscission, we conducted an RNA-seq analysis of petals from position +3 flowers collected from the WT and *snac* (Supplementary Data 8). Because one of the SNAC-A transcription factors, ANAC032, acts as an activator based on transient reporter assays[45,73,80,81], we focused on down-regulated genes in the *snac* mutant compared to the WT and identified 2,615 such genes (Fig. 6d and Supplementary Data 9). We verified the down-regulation of several selected genes by RT–qPCR (Supplementary Fig. 28). Of the 2,615 down-regulated genes, 415 were directly bound by ANAC102 (Fig. 6d and Supplementary Data 10). GO-term enrichment analysis of these targets implicated them in 'response to acid chemical', 'response to hormone', 'regulation of gene expression' and 'regulation of cellular process' such as 'autophagy' and 'cell death'[41,42] (Fig. 6e). REVIGO analysis revealed that the GO terms can be classified into four main groups: 'stress response', 'cellular response', 'development' and 'transport' (Fig. 6f), hinting at previously underappreciated aspects of ANAC102 function.

We also compared the differentially expressed genes in the *dad1* and *snac* mutants: 581 genes were down-regulated in both the *dad1* and *snac* mutants (Supplementary Data 11). Notably, *BFN1* was expressed at lower levels in both *dad1* and *snac* petals (Figs. 2l, n, 6g, and Supplementary Fig. 28). Many of these differentially expressed genes have functions assigned to hormone responses, transcriptional regulation, flower development, cellular responses and ROS production. Genes encoding the basic helix-loop-helix (bHLH) transcription factor SPATULA (SPT), the cytokinin-activating enzyme LONELY GUY1 (LOG1), the ABA receptor PYRABACTIN RESISTANCE 1-LIKE6 (PYL6), the sucrose efflux transporter SUGARS WILL EVENTUALLY BE EXPORTED TRANSPORTER 12 (SWEET12), the Class III PEROXIDASE 33 (PRX33) and PEROXIN11D (PEX11D) were also differentially expressed in the mutants compared to the WT (Supplementary Figs. 28, 29)[82–85]. These observations suggest that ANAC102 orchestrates stress responses, cellular responses, development, transport and ROS production by activating these genes during petal abscission.

### ANAC102 activates autophagy-related genes for proper petal abscission

ROS accumulation and cell death often accompany autophagy[86]. Here, we found multiple autophagy-related genes are direct targets of ANAC102. Among 41 autophagy-related genes, 28 were statistically significantly down-regulated in *snac* petals from position +3 flowers compared to the WT (FDR < 0.05) (Fig. 7a). In particular, no autophagy-related gene was up-regulated in the *snac* mutant (Fig. 7a). Autophagy-related genes in *dad1* petals from position +3 flowers were down-regulated, although to a lesser extent than in *snac* (Fig. 7a). We detected significant ANAC102 binding peaks at the *ATG1b*, *ATG1c*,

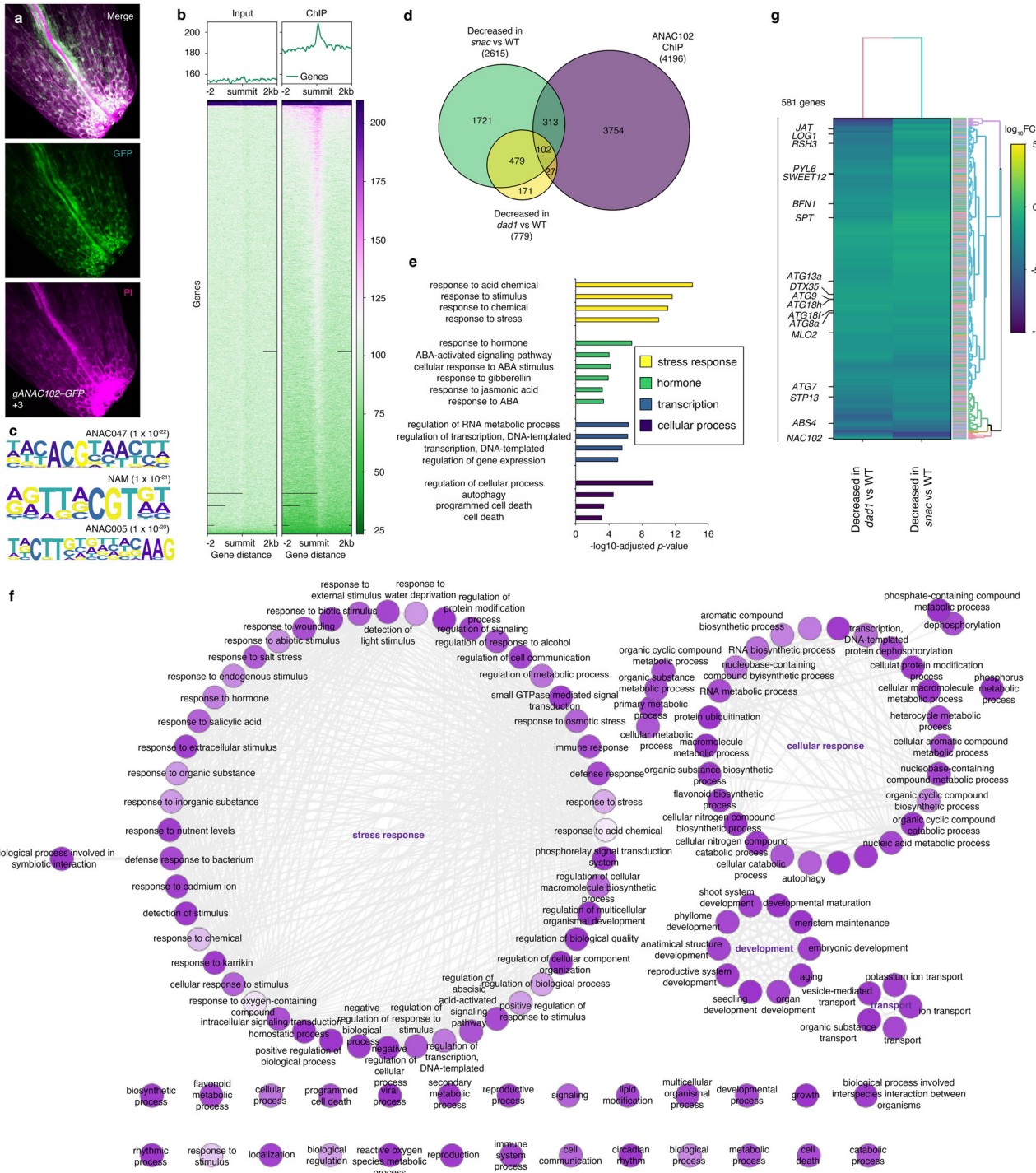

**Fig. 6 | Genome-wide identification of ANAC102-regulated targets during petal abscission. a** gANAC102–GFP accumulation in petal bases from position +3 flowers. Scale bars = 100 μm. Top, merge; middle, GFP; bottom, propidium iodide (PI). **b** ANAC102 input and ChIP data. Left, Input; Right, ChIP. Upper panels show the average profile around the detected peak. Lower panels show read density heatmaps around each detected peak. **c** ANAC102 *cis*-motifs identified by de novo motif analysis under each ANAC102 peak summit. ANAC047 and NAM belong to the SNAC family, as does ANAC102. *p*-values obtained from HOMER are shown. **d** Venn diagram showing the overlap between direct ANAC102 targets (purple), down-regulated genes in the *snac* mutant (green), and down-regulated genes in the *dad1* mutant (yellow). **e** GO-term enrichment analysis using ANAC102-bound genes and down-regulated genes in the *snac* mutant. Selected GO terms related to stress response, plant hormones, transcription and cellular process are shown with their −Log$_{10}$ adjusted *p*-value. The *p*-values for GO analysis are obtained by converting the Z-scores based on a two-tailed Z-test. **f** Interactive graph view of GO terms generated by REVIGO. The node radius indicates the specificity, and the color shading corresponds to the *p*-values. The *p*-values for GO analysis are obtained by converting the Z-scores based on a two-tailed Z-test. Highly similar GO terms are linked by edges, with edge thickness indicating the degree of similarity. **g** Hierarchical clustering of 581 genes during petal abscission. The heatmap shows the Log$_{10}$ fold-change of differentially expressed genes based on the transcriptome data in **d**.

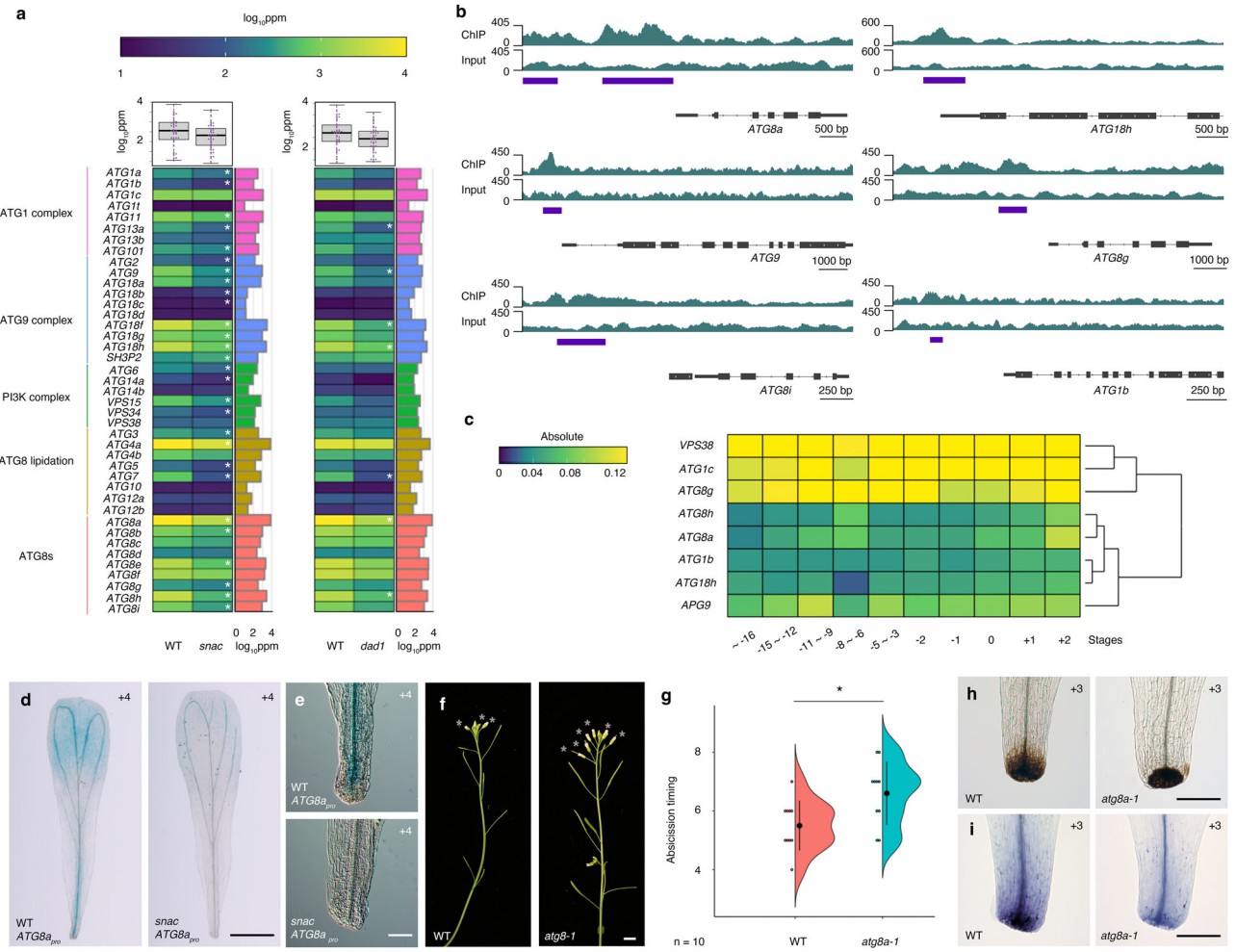

**Fig. 7 | Reduced autophagic activity causes delayed petal abscission. a** *ATG* gene expression in WT, *snac* and *dad1* petals from position +3 flowers (*n* = 3 independent experiments). The heatmap shows the Log₁₀(ppm) genes based on transcriptome data in the mutants relative to the WT. The boxplots above the heatmap show the distribution of expression changes relative to the WT. Boxes encompass the inter-quartiles (25th–75th percentiles), horizontal lines are means, and range bars show the maximum and minimum values excluding outliers. Asterisks indicate statistically significant differences between the WT and mutants based on FDR. The colored bars to the right of the heatmap show expression levels in the WT. **b** IGV browser view of ANAC102 ChIP and input signals at *ATG8a, ATG9, ATG8i, ATG18h, ATG8g* and *ATG1b* loci. Purple horizontal bars indicate a statistically significant difference between ChIP and input signals. The gene models are shown as black bars (coding regions) and lines (non-coding regions) at the bottom of each panel.

**c** Heatmap showing the absolute expression values of eight *ATG* genes based on public transcriptome data. **d** *GUS* reporter staining pattern for the *ATG8a* promoter in WT (left) or *snac* mutant (right) during petal abscission. Scale bar = 1 mm. **e** Zoomed-in views of GUS staining for the *ATG8a* promoter in the WT (top) and the *snac* mutant (bottom) during petal abscission. Scale bar = 100 μm. **f** Side view of WT and *atg8a-1* inflorescences. Open flowers with petals are indicated by asterisks. Scale bar = 1 cm. **g** Quantification of the timing of abscission, shown as individual data points (left) and violin plots (right) for each genotype. Black dots and vertical lines indicate mean and SD, respectively. *n* = 10. One-tailed Student's *t* test, *$p$ = 0.01. **h** DAB staining of WT and *atg8a-1* petals from position +3 flowers. Scale bars = 100 μm. **i** Trypan blue staining of WT and *atg8a-1* petals of position +3 flowers. Scale bar = 100 μm.

*ATG9, VACUOLAR SORTING RECEPTOR38* (*VPS38*), *ATG8a, ATG8g, ATG18h* and *ATG8i* loci (Fig. 6b and Supplementary Fig. 19). *ATG1b, ATG9, ATG8a, ATG8g, ATG18h* and *ATG8i* loci were present in both the DEG list based on RNA-seq and ChIP-seq datasets, and have ANAC102 binding peaks upstream of their transcription start sites (Fig. 7a, b). We extracted the temporal expression pattern of *ATG* genes from the TRAVA database and defined two different clusters. *VPS38, ATG1c* and *ATG8g* were uniformly expressed in petals, suggesting that these genes may contribute to a basal level of autophagy in petals (Fig. 7c). *ATG8a, ATG8h, ATG1b, ATG18* and *APG9* showed a gradual up-regulation across the time course, suggesting a role in petal abscission (Fig. 7c).

Most examples of *ATG8* promoter activity have been examined in shoots and roots[87], with nothing known about expression in petals, prompting us to generate transgenic lines carrying a *ATG8a_pro:GUS* reporter construct. In the WT, we detected strong GUS staining in the petal base, vasculature and blade from position +3 flowers (Fig. 7d, e).

When this same reporter construct was introduced into the *snac* mutant, GUS staining was much weaker (Fig. 7d, e). The overlapping expression pattern of *ANAC102* and *ATG8a* in petal bases and the lower *ATG8a* expression in *snac* are consistent with a possible role for ATG8a downstream of ANAC102 during petal abscission.

To assess the role of autophagy in petal abscission, we characterized the *atg8a* mutant, along with the known autophagy-deficient mutants *atg5* and *atg7*. Similar to ATG8[27,88], ATG5 and ATG7 are both essential for autophagosome formation. Compared to the WT, the timing of petal abscission was statistically significantly delayed in the three *atg* mutants (Fig. 7f, g and Supplementary Figs. 31, 32). In addition, all three *atg* had weaker DAB and trypan blue staining in petals from position +3 flowers compared to the WT (Fig. 7h, i and Supplementary Figs. 31, 32). In summary, these analyses reveal that functional autophagy is required for petal abscission.

## Local autophagy activity at the petal base promotes petal abscission

The *atg* mutants are delayed in petal abscission, but this could be a secondary effect of the pleiotropic phenotypes of these mutants. To directly link petal abscission and autophagy, we manipulated autophagy activity specifically at the petal base and examined the timing of petal abscission. To this end, we expressed *mVENUS–ATG7* from the *ANAC102* promoter in *atg7* petals. In the transgenic plants, we observed dot-shaped mVENUS–ATG7 signals at the base of petals from position +3 flowers (Fig. 8a, b). Expression of *mVENUS–ATG7* rescued the defect in petal abscission of the *atg7* mutant, indicating the fusion construct is functional (Fig. 8c, d and Supplementary Fig. 33). In agreement with this result, the weaker DAB and trypan blue staining intensities seen in petals at position +3 flowers in *atg7* were partially restored by *mVENUS–ATG7* expression (Fig. 8e, f). Thus, local autophagy at the petal base promotes petal abscission.

In a parallel approach, we exploited an artificial microRNA (amiR) targeting *ATG5* or *ATG7* in petal bases, the individual loss of function of which results in delayed petal abscission[89]. We explored the effects of the amiR on its engineered mRNA target by assaying ATG5-GFP accumulation in the *atg5-1* mutant background. In *atg5-1 gATG5–GFP* plants, we observed dot-shaped ATG5–GFP signals at the base of petals (Fig. 8g, h). Additionally, we noticed a weak but uniform ATG5–GFP signal in the cytoplasm of petal base cells. When *amiRATG5* is expressed at the base of petals, ATG5–GFP fluorescence was abolished (Fig. 8g, h). Plants expressing *amiRATG5* and *amiRATG7* had delayed petal abscission (Fig. 8i, j and Supplementary Figs. 33, 34), and had weaker DAB and trypan blue staining in petals at position +3 flowers (Fig. 8l, m).

## Autophagy is activated at the base of petals for subsequent abscission

Toward closely monitoring autophagy dynamics during petal abscission, we used the established autophagosome-marker line $35S_{pro}:GFP–ATG8a$[27,90], which ubiquitously accumulates GFP-tagged ATG8a. This transgenic line allowed us to monitor the vacuolar delivery and subsequent breakdown of GFP–ATG8a as a readout of autophagy function (Fig. 9a–e). In WT position +1 flowers, we detected broad GFP fluorescence in the petal bases, suggesting that autophagic activity is low at this stage (Fig. 9a). Higher magnification of petal-base cells revealed GFP–ATG8a[87] fluorescence is associated with ring-shaped spots, which are typical of autophagosomes (Fig. 9c, d). GFP–ATG8a fluorescence started to reduce, as did the number and size of ring-shaped GFP signals, in WT petals from position +3 flowers, suggesting higher autophagic activity (Fig. 9a, c, e). In WT petals at position +5, GFP fluorescence had largely disappeared (Fig. 9a).

The number of GFP–ATG8a foci in *coi1* petal bases was largely comparable to that of the WT at position +1 (Fig. 9a, b). Although autophagosome-like spots started to decrease in abundance in WT petals at position +3, a similar phenomenon was not observed in *coi1* petal bases at position +3, suggesting that autophagic activity is low at this stage in plants unable to signal JA (Fig. 9a, b). Notably, we often observed small autophagosome-like GFP spots in the *coi1* mutant. In addition, we did not observe the typical ring-shaped spots seen in the WT in *coi1*, which accumulated more like dot-like spots (Fig. 9c, d), suggesting that JA may contribute to autophagosome maturation, as well as autophagic flow. By position +5, GFP fluorescence began to decrease in *coi1* petals (Fig. 9d–f) and had disappeared by position +7 (Fig. 9b). We conclude that delayed autophagic activity correlates with timing of petal abscission in JA-deficient mutants.

When proper autophagic flow occurs, GFP–ATG8a is delivered to the vacuole, where it is degraded by vacuolar proteases[91]. After lysis of the autophagic-body membrane, the contents are exposed to vacuolar hydrolases. Since we observed more autophagic bodies in JA-related and *snac* petal-base cells compared to the WT, we monitored

GFP–ATG8a degradation and release of free GFP in the vacuole by western blot with protein extracts from petals. In the WT at petal position +2 flowers, we detected intact GFP–ATG8a, in addition to a smaller protein with a molecular mass consistent with free GFP (Fig. 9f), suggesting the cleavage of GFP from GFP–ATG8a or the degradation of ATG8a from GFP–ATG8. The abundance of intact GFP–ATG8a progressively declined from petals at positions +3 to +4 before almost disappearing by position +5 (Fig. 9f). As WT petals grew, the signal of free GFP also decreased but remained detectable by position +5, suggesting that proper autophagy for petal abscission occurs by this position in the WT (Fig. 9f, g).

In *coi1* petals at position +2, we detected intact GFP–ATG8a at roughly the same levels as in the WT (Fig. 9f). GFP–ATG8a stayed constant from position +3 to +5 in the *coi1* petals (Fig. 9f). Furthermore, although we also detected a band of a size consistent with free GFP, its abundance stayed at similar levels up to position +5 petals (Fig. 9f). The degradation of the GFP–ATG8a fusion protein in the WT was faster than in the *coi1* mutant (Fig. 9f, g). Consistent with this result, the ratio of free GFP to GFP–ATG8a in WT petals at later stages was higher than in the *coi1* mutant, suggesting diminished breakdown of cellular components in this JA-deficient mutant (Fig. 9f, g). Taken together, autophagosome maturation, vacuolar delivery, and subsequent breakdown are regulated by the JA pathway in petals.

To confirm the effect of inhibiting autophagic flux on petal abscission, we treated plants with concanamycin A, which inhibits vacuolar type H+-ATPase activity and causes autophagic bodies to accumulate in vacuoles[92]. In the WT, GFP–ATG8a accumulated broadly at the base of petals at position +1 and gradually degraded as developmentally linked autophagy progressed (Supplementary Fig. 35a). This degradation was compromised in concanamycin A-treated petal bases at position +2 and +3 and did not affect autophagosome shape (Supplementary Fig. 35a–e). Quantification of autophagy activity using a flux assay supported this finding (Supplementary Fig. 35f, g), confirming that genetic and pharmacological perturbations have similar effects on autophagic flux at the base of petals, at least in part.

## Discussion

### JA controls timing of petal abscission

JA-related mutants are male-sterile, underscoring the essential role played by this hormone in flower development[93]. Our findings confirm that proper petal-abscission timing is also a function of JA signaling to induce petal senescence and is under AG regulation. Indeed, we observed delayed abscission in the mutants *ag*, *dad1*, *aos* and *coi1* compared to WT plants that was negatively correlated with ROS accumulation shortly before abscission (position +3 flowers, Fig. 1b–g). The cells distal to the petal base in position +3 flowers in *ag* and *dad1* also had distinct cellular morphology and homeostasis, suggesting that AG and JA promote petal abscission by maintaining cellular homeostasis at the petal base. In particular, we observed numerous small vesicles within vacuoles in the *ag* and JA-related mutants, suggesting that various vacuolar-trafficking pathways at the base of petals might be perturbed by these mutations. Our results indicate that AG and DAD1 positively regulate petal abscission and that JA signaling is necessary for proper petal-abscission timing. Because the *dad1* mutant phenotype was intermediate from that of *aos* or *coi1* mutants, we hypothesize that redundant factors contribute to petal abscission, such as *DAD1* homologs[30]. Additionally, JA signaling acts during phase two of floral-organ abscission[18], during which AG may be an upstream regulator by regulating JA biosynthesis through DAD1 to bring about the necessary JA feedback loop for abscission-zone cells to acquire abscission competence. Finally, the lower ROS accumulation in petals delayed abscission in mutants, confirming previous observations that floral-organ abscission and senescence are tightly linked[18].

Importantly, exogenous JA application to *ag* and *dad1* plants rescued their delay in petal abscission, which was preceded by ROS

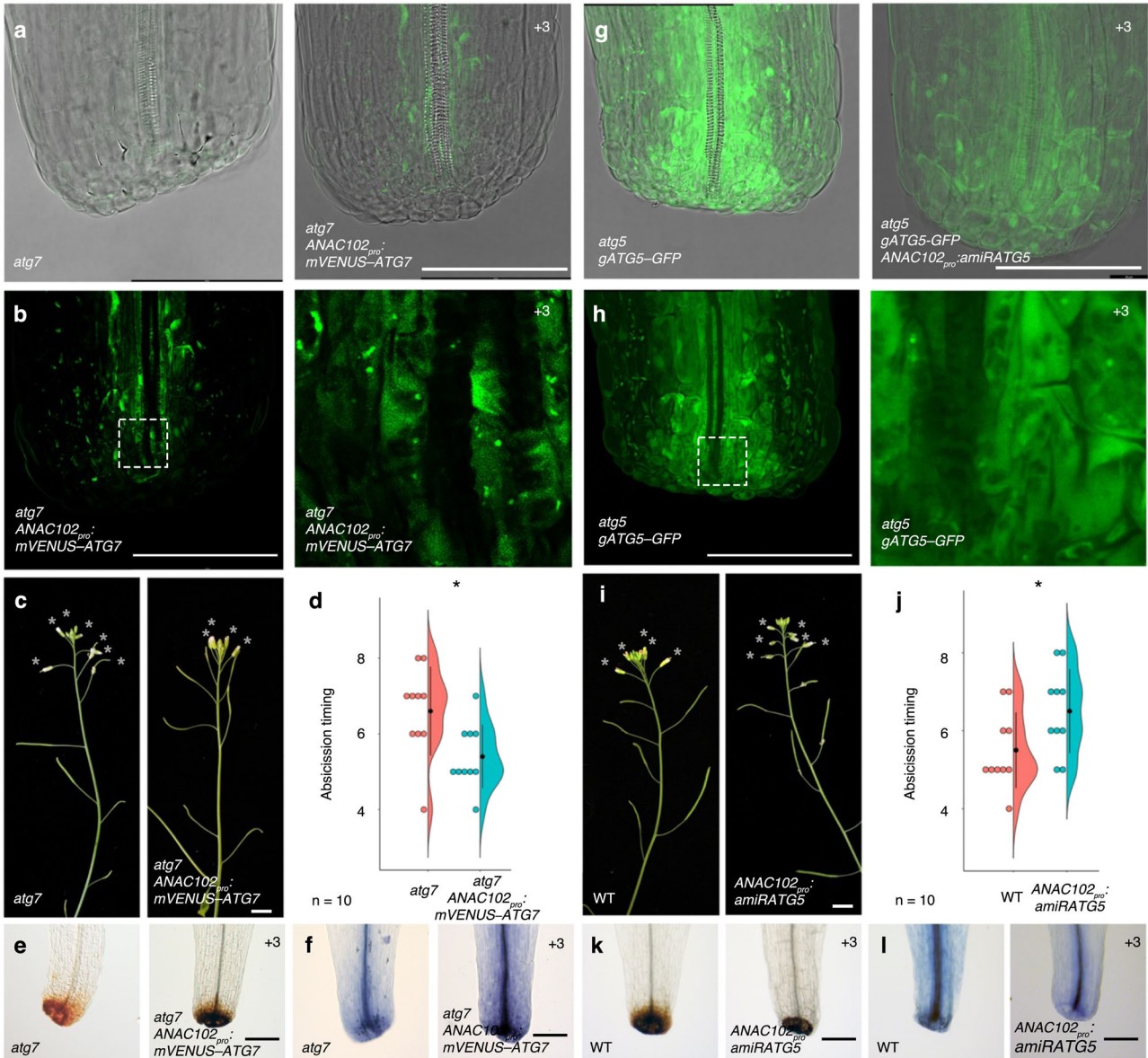

**Fig. 8 | Local changes in autophagic activity affect the timing of petal abscission. a** Spatiotemporal observation of VENUS in petal bases of *atg7-2* (left) and *atg7-2 ANAC102pro:mVENUS−ATG7* (right) by confocal microscopy. Scale bars = 100 μm. **b** Higher magnification images of VENUS in petal bases of *atg7-2 ANAC102pro:mVENUS−ATG7* by confocal microscopy. Image in right is magnified image of the boxed region in left. Scale bars = 100 μm. **c** Side view of *atg7-2* and *atg7-2 ANAC102pro:mVENUS−ATG7* inflorescences. Open flowers with petals are indicated by asterisks. Scale bar = 1 cm. **d** Quantification of the timing of abscission, shown as individual data points (left) and violin plots (right) for each genotype. Black dots and vertical lines indicate mean and SD, respectively. *n* = 10. One-tailed Student's *t* test, *p = 9.1 × 10⁻³. **e** DAB staining of *atg7-2* and *atg7-2 ANAC102pro:mVENUS−ATG7* petals of position +3 flowers. Scale bars = 100 μm. **f** Trypan blue staining of *atg7-2* and *atg7-2 ANAC102pro:mVENUS−ATG7* petals of

position +3 flowers. Scale bar = 100 μm. **g** Spatiotemporal observation of GFP in petal bases of *atg5-1 gATG5−GFP* (left) and *atg5-1 gATG5−GFP ANAC102pro:amiRATG5* (right) by confocal microscopy. Scale bars = 100 μm. **h** Higher magnification images of GFP in petal bases of *atg5-1 gATG5−GFP* by confocal microscopy. Image in right is magnified image of the boxed region in left. Scale bars = 100 μm. **i** Side view of WT (left) and *ANAC102pro:amiRATG5* (right) inflorescences. Open flowers with petals are indicated by asterisks. Scale bar = 1 cm. **j** Quantification of the timing of abscission, shown as individual data points (left) and violin plots (right) for each genotype. Black dots and vertical lines indicate mean and SD, respectively. *n* = 10. One-tailed Student's *t* test, *p = 0.02. **k** DAB staining of WT (left) and *ANAC102pro:amiRATG5* (right) petals of position +3 flowers. Scale bars = 100 μm. **l** Trypan blue staining of WT (left) and *ANAC102pro:amiRATG5* (right) petals of position +3 flowers. Scale bar = 100 μm.

accumulation in the petal bases of position +3 flowers. These results highlight how AG and DAD1 play a role in coordinating JA signaling in properly timing petal abscission. The JA sensor JAZ9−VENUS revealed that petal bases of position-0 flowers are sites of active JA signaling, consistent with a model of JA biosynthesis (*DAD1*, *AOS*, and *OPR3* expression) occurring in the stamens for proper pollen maturation and anther dehiscence, and coordination of petal development and flower opening by diffusion or active transport of JA to petals[30,94,95].

Because JA is also involved in stamen abscission, the exact mechanism by which JA diffuses or is transported to direct abscission between different organs needs to be resolved. This specific period of JA signaling activity was accompanied by programmed cell death, as heralded by *BFN1* expression at the petal base of position-0 flowers. *BFN1* was undetected in *ag* and *dad1* mutants at the base of position-0 flower petals but was restored to WT levels upon JA treatment.

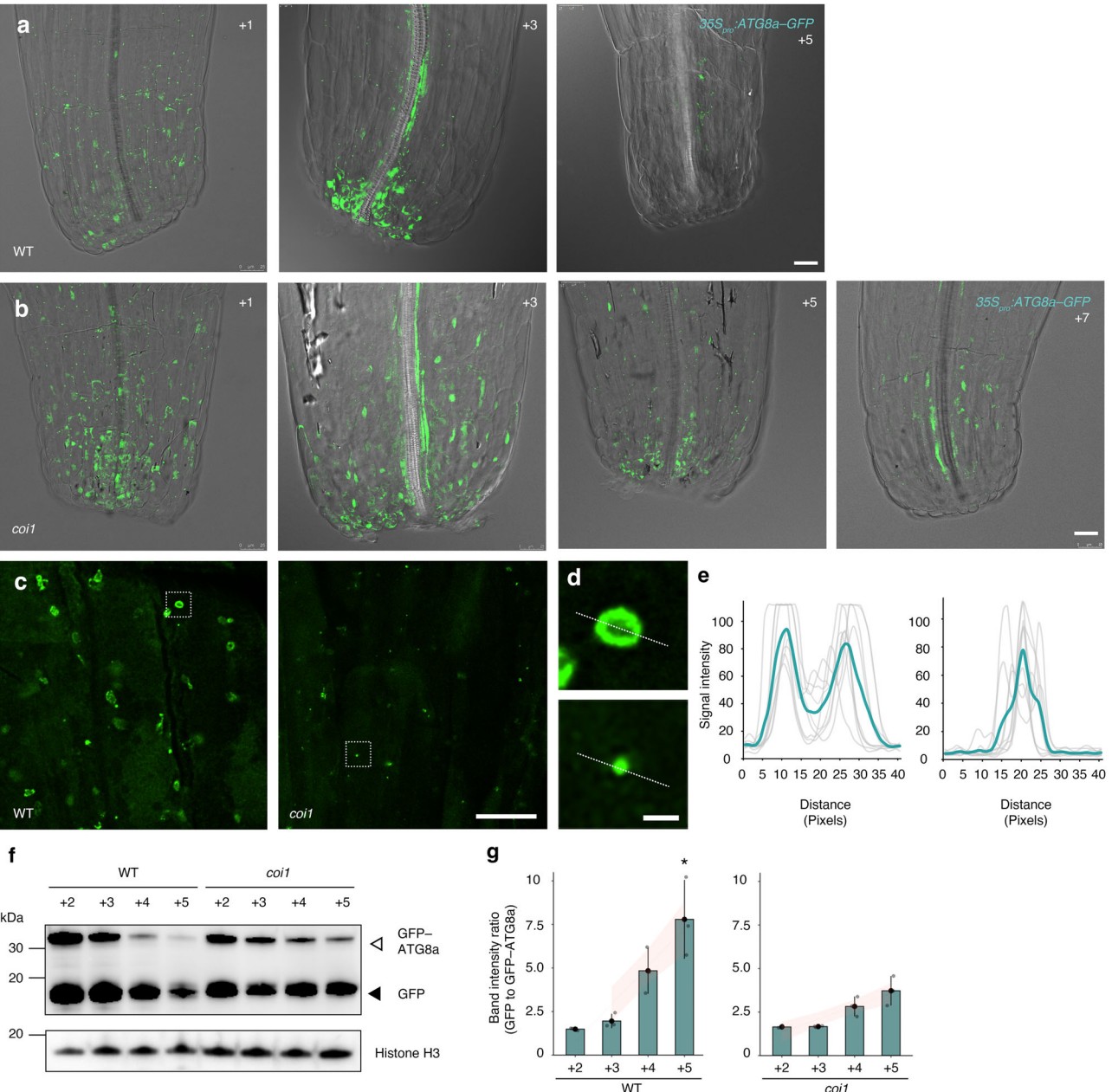

**Fig. 9 | Autophagosome degradation at the petal base during petal abscission is delayed in JA mutants.** Spatiotemporal observation of the autophagosome marker GFP–ATG8a in petal bases of the WT (**a**) and *coi1* (**b**) by confocal microscopy. Scale bars = 25 μm. **c**, **d** Higher magnification images of *35S_pro:GFP–ATG8a* in petal bases of the WT (left) and *coi1* (right) by confocal microscopy. Images in **d** are magnified images of the boxed regions in **c**. The punctate structures labeled by green fluorescence from the cleavage of GFP–ATG8a indicate autophagosome-related structures. Scale bars in **c** = 10 μm. Scale bars in **d** = 1 μm. Numbers in the right corners indicate flower position. **e** GFP–ATG8a fluorescence-intensity profile in WT and *coi1* autophagosome-like structures. *n* = 9. Each gray line indicates an individual trace.

The green line shows mean data. **f** Immunoblot analysis showing the processing of GFP–ATG8a in the WT and the *coi1* mutant during petal abscission. Crude protein extracts from petals at the indicated positions were subjected to SDS–PAGE and immunoblot analysis with anti-GFP antibodies. Arrows and arrowheads to the right side of immunoblot image indicate GFP–ATG8a and free GFP, respectively. Histone H3 served as a loading control. **g** Quantification of the GFP/GFP–ATG8a ratio during petal abscission by densitometric scans of the immunoblots shown in **g**. Values are means ± SE (*n* = 3 independent experiments). Asterisks indicate statistically significant differences between the WT and the *coi1* mutant based on two-tailed Student's *t* test *p* = 0.04.

## A JA-mediated chromatin switch enables abscission gene expression in petals

MYC2 is the gatekeeper that relays the JA signal across transcriptional networks to orchestrate a cascade of downstream responses by interacting with TPL or MED25 for proper reprogramming of the chromatin environment[14,65]. Here, we identified a transcriptional network starting with MYC2 and its interactors that transmits the JA signal essential for petal-abscission timing. We focused on direct downstream targets of MYC2 and MED25 based on their high expression in

position +3 petals: the NAC transcription-factor gene *ANAC102*[96], the hexose-transporter gene *STP13* that functions during programmed cell death[44], the transcription-factor gene *RAV1* that promotes leaf senescence[97] and the blue-copper endomembrane gene *SAG14* that regulates dark-induced leaf senescence[98]. These four proteins are associated with programmed cell death and senescence and form part of the regulatory module underlying AG–DAD1-mediated JA-induced programmed cell death in petal abscission. We showed that accessibility to the local chromatin environment was crucial to their timely

activation in response to JA signaling. We propose that MYC2 serves as a convergence point for controlling chromatin accessibility as a function of its interacting partner: limited chromatin access via MYC2 interactions with the repressive TPL–HDAC module during low JA signal output or greater access by recruiting MED25 to induce targets when an activating JA signal is perceived. Variability in the regulatory components of this MYC2 switch suggests that it can be fine-tuned in response to specific signals, thus placing the MYC2 switch at the core of a signaling hub to coordinate the transcriptional response to JA signaling and set the timing of petal abscission.

## ANAC102 is a key JA signaling component for correct petal-abscission timing

NAC transcription factors responding to environmental stresses were grouped into SNAC subclasses giving two distinct phylogenetic groups: SNAC-A and SNAC-B[45]. Whereas SNAC-A members were initially thought to be associated with abiotic stress responses and SNAC-B members were involved in senescence responses[45], some functional overlap has been reported[45]. Here, we defined a set of ANAC102 direct downstream targets during petal abscission and their putative functions. Our results suggest that ANAC102 plays a key role in the JA-mediated petal-abscission pathway, as ectopic expression of *ANAC102* can partially rescue petal-abscission defects in *dad1* and *ANAC102* is expressed in petal bases and the vasculature from position +3 flowers. These findings demonstrate that ANAC102 is a major regulatory node in AG–JA-dependent petal-abscission timing. Our results also indicate that ANAC102 coordinates the transcriptional response to environmental and endogenous cues, possibly to facilitate the intracellular shift to senescence, prior to floral-organ abscission. The activation of *SPT*, *LOG1*, *PYL6* and *SWEET12* indicate further interactions with other hormone and metabolic pathways.

## ANAC102 times local autophagy for petal abscission

Because flower longevity is critical for ornamental plants, cellular dissection of petals has been conducted in many plant species[4,28,99]. Although dynamic changes in the number of vesicles and cytoplasmic components in the vacuole suggest a key role for autophagy during petal development, it remained unknown how autophagy machinery is activated during petal senescence and abscission. In this study, we conclude that JA–ANAC-induced *ATG* genes control petal-cell homeostasis via autophagy and affect subsequent petal abscission.

Because of the expression pattern of ANAC102, the timing and location of *ATG* expression should also be tightly regulated during petal abscission. Indeed, five out of eight ANAC102 direct target genes reached their expression peaks at later stages of petal development, suggesting that ANAC102 controls *ATG* induction (Fig. 7c). Although ANAC102 accumulation preceded that of *ATG8a* expression, *ATG8a* was strongly expressed in the petal base and blade. Because *ATG8a* expression in the blade was also compromised in the *snac* mutant, SNACs other than ANAC102 might also induce *ATG8a* expression in this tissue. How the SNAC family functions redundantly to specify the spatial *ATG8a* expression domain needs to be resolved.

Consistent with the role of autophagy in organelle rearrangement associated with differentiation, autophagy-deficient mutants are delayed in petal abscission. The weak petal-abscission phenotype in autophagy-deficient mutants relative to JA-deficient mutants suggests that other redundant ATG proteins and/or additional pathways participate in petal abscission. ATG proteins mainly control autophagosome maturation and subsequent vacuolar delivery. In JA-deficient mutants, autophagosome maturation was compromised, as was the breakdown of cellular components in the vacuole as assayed by GFP cleavage (Fig. 9). Similar to JA-deficient mutants, concanamycin A-treated plants also showed impairment in breakdown of cellular components, suggesting that the JA pathway also affected vacuole

breakdown by an unknown mechanism. Notably, senescence-specific cysteine proteases, such as SAGs, often localize to the vacuolar compartment during leaf senescence[100], and *SAG14* is a mutual direct target of MYC2 and MED25 (Fig. 4c). The role of each MYC2 and MED25-shared target needs to be examined.

In summary, our study reveals a molecular mechanism behind spatiotemporal regulation of petal abscission and a previously undescribed role for developmentally regulated autophagy by a JA–NAC module acting in flowers (Fig. 10). A classic role for JA during plant growth and development is organ senescence[101]. Notably, NACs have been implicated in petal senescence, as evidenced by the identification of the NAC protein EPHEMERAL1 in Japanese morning glory[22]. Furthermore, autophagic changes are seen in petals in many angiosperms[4,28,99], suggesting that JA–NAC-mediated autophagy may be conserved in angiosperms.

# Methods

## Plant materials and growth condition

The Arabidopsis (*Arabidopsis thaliana*) accession Columbia-0 (Col-0) was used in this study. The mutants *ag* (SALK_014999)[102], *dad1-3* (SALK_037352)[103], *aos*[103], *coi1-51*[104], *myc2 myc3 myc4*[104], *med25-4*[56], the *snac* septuple mutant[45], *atg8a-1* (SALK_012133), *atg5-1*[105] and *atg7-2* (GK-655B06)[27] and the transgenic lines *DAD1_{pro}:GUS*[30], *35S_{pro}:JAZ9–VENUS*[34], *BFN1_{pro}:nGFP*[106], *jin1-8 MYC2_{pro}:MYC2–FLAG*[107], *35S_{pro}:TPL–GFP*[108], *HDA6_{pro}:HDA6–GFP*[109], *HDA19_{pro}:HDA19–GFP*[110], *35S_{pro}:MED25–GFP*[56], *atg5-1 gATG5–GFP*[27], and *35S_{pro}:GFP–ATG8a*[90] were previously described. To analyze floral tissues, plants were grown on soil consisting of vermiculite and Metro-Mix mixed in a 2:1 ratio. HYPONeX fertilizer diluted 1,000x (HYPONeX JAPAN) was given every 5 days until analysis. For selection based on antibiotic resistance or fluorescence observation for genotyping, seeds were sown on half-strength Murashige and Skoog (MS) plates containing 0.8% (w/v) agar. Seedlings and plants were grown at 22 °C under continuous light conditions at 200 μmol cm$^{-2}$ s$^{-1}$ light intensity. Primers used for genotyping are listed in Supplementary Data 12.

## Plasmid construction and plant transformation

To generate the *ANAC102* and *ATG8a* promoter constructs, the genomic region upstream of each gene was amplified by PrimeSTAR GXL DNA polymerase (TaKaRa) by gradient PCR, gel-purified with a gel/PCR extraction kit (NIPPON Genetics) using Col-0 template genome DNA, and cloned into pENTR/D-TOPO (Thermo Fisher Scientific) via Gateway cloning. The sequence of the insert was confirmed by Sanger sequencing. The resulting construct was used as a template to introduce mutations into the *ANAC102* promoter. Site-directed mutagenesis was conducted using a site-directed mutagenesis kit (NEB) following the manufacturer's instructions. Primers used for promoter cloning are listed in Supplementary Data 12. The promoter fragments were recombined into pBGWSF7.0 GUS–GFP vector via LR reaction using LR Clonase II mix (Thermo Fisher Scientific)[111]. After checking the junction between promoter insertion and *GUS–GFP*, the plasmids were transformed into Agrobacterium (*Agrobacterium tumefaciens*) strain GV3101[112]. Positive Agrobacterium colonies were used for Arabidopsis transformation by the floral dip method[112].

To express *ANAC102* in a petal-specific manner, the promoter sequence of *InMYB1* (from the Japanese morning glory [*Ipomoea nil*] *MYB1* gene) was PCR amplified from the *InMYB1_332-121 bp_TATA_Ω* plasmid using gene-specific primers[75,76]. The *ANAC102* coding sequence was also amplified from Arabidopsis Col-0 cDNA using specific primers. Fragment size was confirmed by gel electrophoresis. The resulting gel-purified *InMYB1* and *ANAC102* fragments were inserted into the pENTR2B vector with NEBuilder HiFi DNA Assembly Master Mix (New England BioLabs). After confirmation by Sanger sequencing, the *pInMYB1_332-121 ×3_TATA_Ω:ANAC102* cassette was recombined into the pGWB533 binary vector by LR reaction using LR Clonase II

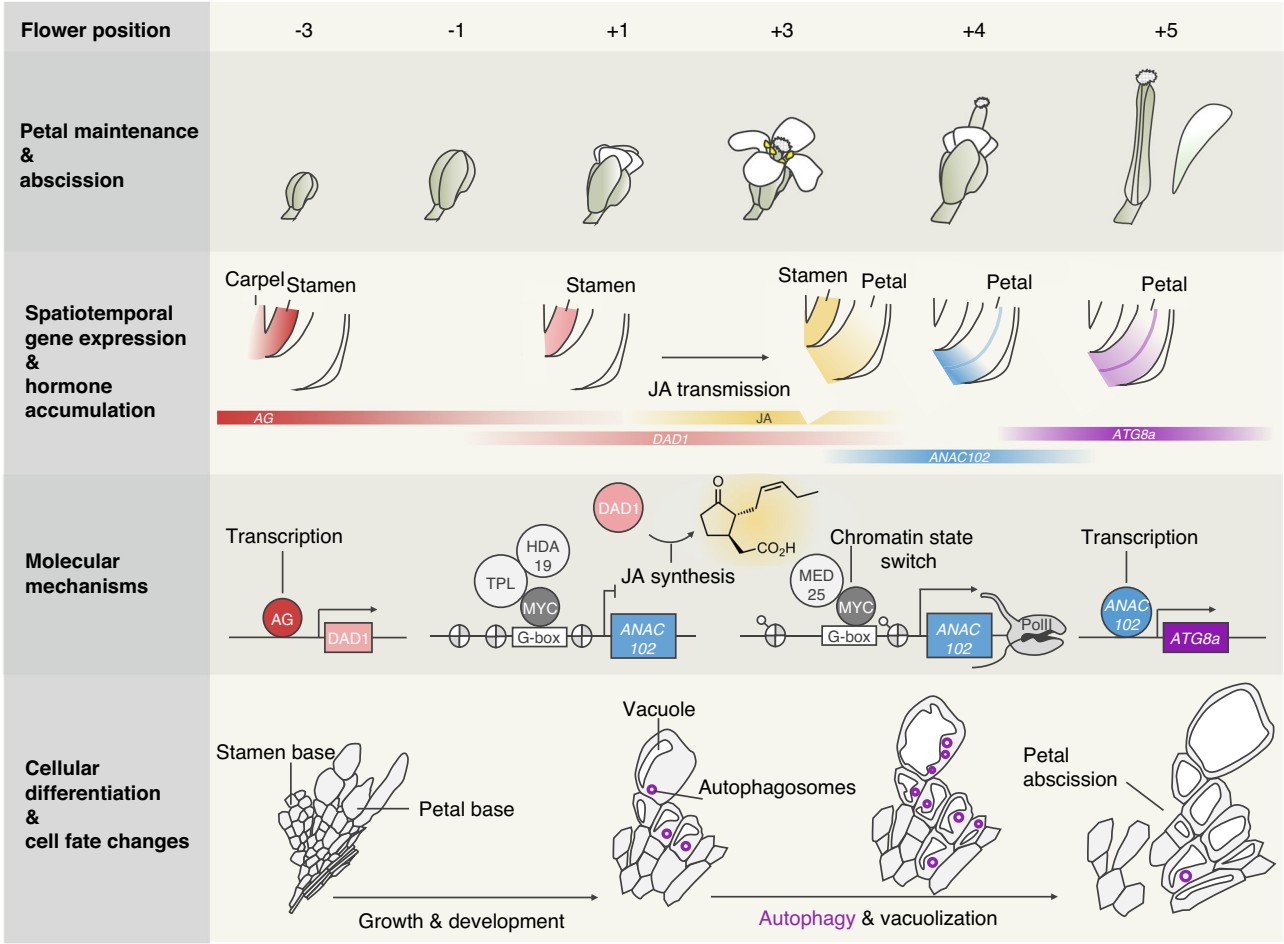

**Fig. 10 | Model of local cell-fate decisions arising from the interplay between JA and autophagy during petal abscission.** Before induction of JA biosynthesis by AG, JAZ repressors accumulate at the base of petals to block MYC activity, leading to lower ROS levels and cell death. JA biosynthesis in stamen filaments induced by AG triggers JAZ degradation at the base of petals, allowing MYC transcription factors to recruit MED25, deposit H3K9ac, and increase chromatin accessibility for key downstream targets, such as ANAC102. Subsequently, ANAC102 accumulates at the base of petals just prior to petal abscission and promotes proper autophagy through transcriptional activation of ATG genes.

(Thermo Fisher Scientific). Primer sequences are listed in Supplementary Data 12. The plasmid was transformed into Agrobacterium strain GV3101. Positive colonies were used for Arabidopsis transformation by the floral dip method. One day after the *dad1* mutant plants were transformed, flower buds were sprayed with 500 mM methyl jasmonate (Wako) containing 0.5% (v/v) Tween 20 to achieve fertilization.

To observe DAD1 protein accumulation, a *DAD1* genomic fragment was PCR amplified from genomic DNA using NEBuilder HiFi DNA Assembly Master Mix (New England BioLabs) and a gene-specific primer set. The resulting genomic fragment was subcloned into the pENTR2B entry vector using NEBuilder HiFi DNA Assembly Master Mix (New England BioLabs). After confirmation by Sanger sequencing, the genomic *DAD1* fragment was recombined into the pGWB540 destination vector. Primer sequences are listed in Supplementary Data 12. The plasmid was transformed into Agrobacterium strain GV3101. Positive colonies were used for Arabidopsis transformation by the floral dip method[112]. One day after the *dad1* mutant plants were transformed, flower buds were sprayed with 500 mM methyl jasmonate (Wako) containing 0.5% (v/v) Tween 20 to achieve fertilization.

To express *mVENUS–ATG7* under the control of the *ANAC102* promoter, the promoter region was PCR amplified from the *ANAC102pro:GUS–GFP* construct (in the pBGWSF7.0 backbone) as a template using PrimeSTAR GXL DNA polymerase (TaKaRa) and a gene-

specific primer set. Prior to ligation, 3′ A overhangs were added to gel-purified PCR fragments at their 3′ ends using dA-overhang Reaction Mix (Nippon Gene). The resulting DNA fragments were TA-cloned into pENTR 5′-TOPO. The *mVENUS–ATG7* DNA sequence was amplified using gene-specific primer sets for *VENUS* and *ATG7* with PrimeSTAR GXL DNA Polymerase (TaKaRa). A GGSG linker was added at the junction between mVENUS and ATG7. The *mVENUS–ATG7* DNA fragment was introduced into pENTR/D-TOPO (Thermo Fisher Scientific). Using the LR reaction with R4pGWB601[113], 5′-TOPO-ANAC102pro, and pENTR/D-TOPO-*mVENUS–ATG7*, yielding the R4pGWB601-*ANAC102pro:mVENUS–ATG7* plasmid. The plasmid was then transformed into Agrobacterium strain GV3101. Positive colonies were used for Arabidopsis transformation by the floral dip method[112].

To knockdown *ATG5* and *ATG7*, amiRNAs[89] against *ATG5* and *ATG7* were designed using Web MicroRNA Designer (WMD3, http://wmd3.weigelworld.org/cgi-bin/webapp.cgi?page=Home;project=stdwmd) and generated following the recommend protocol. The resulting DNA fragments were subcloned into pENTR/D-TOPO (Thermo Fisher Scientific) via the Gateway system and sequenced. Using the LR reaction with R4pGWB601, 5′-TOPO-ANAC102pro, and pENTR/D-TOPO-*amiR*, the R4pGWB601-*ANAC102pro:amiRATG5* and *ANAC102pro:amiRATG7* plasmids were generated. The plasmid was transformed into Agrobacterium strain GV3101. Positive colonies were used for Arabidopsis transformation by the floral dip method[112].

Antibiotic resistance was used to identify transgenic T1 transformants. Single-copy transgenic lines were determined by examining the segregation ratio of antibiotic resistance among the T2 progeny resulting from self-pollination. Following the identification of T2 lines, they were further propagated to T3 generation.

## Phenotyping of petal abscission

To minimize environmental differences in growth chambers, plants were grown side-by-side at the same density in pots. Five-week-old plants were used for petal phenotyping. To define petal position along the primary inflorescence stem in wild-type, mutant and transgenic plants, the timing of first organ protrusion from floral buds was defined as position 1, as reported previously[29]. In the wild type, position 1 represents the youngest stage-13 flowers, the petals of which are visible from the floral buds. Earlier- or later-formed flowers were assigned minus and plus numbers, respectively. The flower position when all four petals had abscised was scored in both the wild type and mutants. Analysis of variance (ANOVA) with post-hoc Tukey's or Student's $t$-test were conducted to evaluate the statistical significance of the data for multiple and single comparisons, respectively. ANOVA was conducted with an online tool (https://astatsa.com/OneWay_Anova_with_TukeyHSD/). Student's $t$-test was performed using Microsoft Excel (Microsoft 365 16.61). Graphs were generated in R (2021.09.1 Build 372) with the ggplot2 package (3.3.5).

## DAB staining and signal quantification

To detect $H_2O_2$ accumulation in petals, DAB staining was conducted using a Peroxidase Stain DAB Kit (Nacalai)[114]. Petals at position +3 were harvested with forceps and placed into 1.5-mL tubes containing DAB staining solution. The resulting tubes were entirely covered with aluminum foil and shaken orbitally for 1 h at room temperature. After replacing the staining solution with clearing solution (ethanol:acetic acid:glycerol = 3:1:1 v/v/v), petal clearing was conducted by boiling the samples for 5 min in a block heating system (ASTEK). Petals were then mounted on a microscope slide with one or two drops of clearing solution, covered with a cover glass, and imaged using an Axio Scope A microscope (Zeiss) or Stemi DV4 (Zeiss), an AxioCam ERc 5 s camera (Zeiss), and ZEN2 software (Zeiss). The DAB staining area was quantified by ImageJ (National Institutes of Health) using more than 10 petals from different flowers for each genotype. ANOVA with post-hoc Tukey's test was conducted to evaluate the statistical significance of the data in multiple comparisons. ANOVA was conducted as above. Graphs were generated in R (2021.09.1 Build 372) with the ggplot2 package (3.3.5).

## Trypan blue staining

To visualize dead cells, trypan blue staining was conducted[115]. Petals at position +3 were harvested with forceps, placed into 1.5-mL tubes containing 0.5% (v/v) trypan blue solution (Nacalai), and boiled for 5 min. Petals were then washed more than three times. Washed petals were transferred into chloral hydrate solution to make them transparent. Petals were imaged using an Axio Scope A microscope (Zeiss) or Stemi DV4 (Zeiss), an AxioCam ERc 5 s camera (Zeiss), and ZEN2 software (Zeiss). More than 10 petals for each genotype were observed, and representative images are shown.

## Electron microscopy

Flowers of wild-type and mutant petals at the +3 positions were placed in fixation solution (4% [w/v] formaldehyde, 2% [v/v] glutaraldehyde, and 0.05 M cacodylate buffer) at 4 °C overnight. After fixation with 1% (v/v) osumium tetroxide for another 2.5 h at room temperature, the flowers were dehydrated in a graded ethanol series, which was then replaced with EPON812 (TAAB) for sample embedding. Resin blocks were cut to 1-μm thickness with an ultramicrotome (Leica microsystems EM-UC7) using a diamond knife (DIATOME Histo), and serial sections were placed on glass slides. Sections stained with 0.05% (w/v) toluidine blue were examined under an optical microscope (Olympus BX 53 M and DP26 digital camera), and a section in the center of the petal was selected. After staining with 0.4% (w/v) uranyl acetate solution for 12 min and lead citrate solution for 3 min, the sections were coated with osmium coater (HPC-1SW) for 3 s. A field emission scanning electron microscope (FE-SEM, Hitachi High-Tech SU8220, YAG-BSE 5 kV) was used for wide-area imaging. The images were stitched together using Image-Pro software (Media Cybernetics).

## GUS staining

Plant tissue was fixed at room temperature in 90% (v/v) ice-cold acetone for 15 min, rinsed in GUS staining buffer without 5-bromo-4-chloro-3-indolyl-b-D-glucurinde (X-Gluc), and stained with GUS staining solution (50 mM $NaPO_4$, 2 mM $K_4Fe(CN)_6$, 2 mM $K_3Fe(CN)_6$, 2 mM X-Gluc) at 37 °C overnight. The resulting stained tissues were incubated in 70% (v/v) ethanol for at least 1 week. Plants to be directly compared were grown and stained at the same time. Petals were then mounted on a microscope slide with one or two drops of 70% (v/v) ethanol and covered with a cover glass. The resulting petals were imaged using an Axio Scope A microscope (Zeiss) or Stemi DV4 (Zeiss), an AxioCam ERc 5 s camera (Zeiss), and ZEN2 software (Zeiss). More than 10 petals at each position were observed, and representative images are shown.

## Jasmonate and concanamycin A treatment of plants

For JA treatment, 5–10-cm inflorescences were sprayed with 500 μM methyl jasmonate (FUJIFILM Wako Chemicals) and 0.5% (v/v) Tween 20 (FUJIFILM Wako Chemicals). Consistent with previous findings[30], phenotypic rescue was observed in 2 or 3 flowers per plant at 2 days after JA application. This timing-dependent partial rescue of the mutant phenotype can be attributed to differences in the stages of flowers under treatment. The resulting petals were analyzed as described above.

Concanamycin A (BioViotica: BVT-0237) treatment was conducted as reported previously[116]. The distal 5–10-cm inflorescences were placed into a 1.5-mL tube containing half-strength MS medium with 1 μM concanamycin A and incubated overnight under continuous light at 4 °C. The resulting petals were analyzed as described above.

## Confocal microscopy

For GFP observation, plants that were to be directly compared were grown side-by-side at the same density per pot to minimize potential micro-environmental differences in the growth chamber. Petals were harvested in PCR tubes containing 3.3% (w/v) paraformaldehyde (Nacalai tesque) and 0.16 mg/mL propidium iodide (Sigma) dissolved into 1× phosphate-buffered saline (PBS) (Takara) and incubated overnight at 4 °C. The paraformaldehyde solution was replaced with Clear See (FUJIFILM Wako Chemicals) and maintained for about 1 week until the tissues became transparent. The resulting tissues were placed onto a NEO micro slide glass (MATSUNAMI), mounted on a NEO micro cover glass (MATSUNAMI) with one or two drops of water or Clear See, and observed immediately with a confocal laser scanning microscope (SP8 or Stellaris 5: Leica). The same offset and gain settings were used for all plants for which signal was directly compared. More than 10 petals for each genotype and each position were observed, and representative images are shown.

## Analyses of public transcriptome datasets

Three available public transcriptome datasets were used to identify genes that are regulated by JA and JAZ repressors. Published lists of differentially expressed genes from wild-type seedlings treated with 30 μM JA-IIe at two different time points (CRA000278) and the *jaz* decuple mutant (GSE116681) were downloaded from NCBI[16,38].

Common differentially expressed genes in response to JA treatment and in *jaz-D* and *dad1* mutants were defined as high-confidence DAD1 downstream genes and identified with VENNY v.2.1.0 (https://bioinfogp.cnb.csic.es/tools/venny/). The expression of 70 high-confidence DAD1 downstream genes was represented as a heatmap generated in R (2021.09.1 Build 372) using the heatmaply package (1.3.0). The AgriGO v2.0 website (http://systemsbiology.cau.edu.cn/agriGOv2/) was used to perform GO-term enrichment analysis[41,117]. After removal of redundant GO terms by Reduce + VIsual Gene Ontology (REVIGO) (http://revigo.irb.hr), visual representations (interactive graph view and TreeMap) were generated[42]. The resulting Xgmml file was downloaded from the REVIGO website and visualized in Cytoscape3.7.0 (https://cytoscape.org/index.html).

Expression levels of common 25 targets between MYC2 and MED25 were downloaded from the TraVawebsite (http://travadb.org/browse/)[63]. The resulting heatmap was drawn in R (2021.09.1 Build 372) using the superheat package (0.1.0).

## RT−qPCR and RNA-seq analyses
Total RNA was extracted from petals at position +3. For each genotype, 20−40 petals were harvested, immediately frozen in liquid nitrogen, and stored at −80 °C until use. The tissues were ground to a fine powder with small pestles in 1.5-mL tubes. Total RNA extraction was conducted using an RNeasy Plant Mini Kit (Qiagen) following the manufacturer's instructions. Genomic DNA was removed with a RNase-Free DNase set (Qiagen) prior to first-strand cDNA synthesis. The concentration of RNA was determined with a spectrophotometer (IMPLEN NanoPhotometer P-Class).

First-strand cDNA was synthesized with a PrimeScript 1st strand cDNA Synthesis Kit (Takara) using 100 ng of total RNA as starting template. For RT−qPCR, the cycle threshold (Ct) was defined on a LightCycler 480 (Roche) instrument using a FastStart Essential DNA Green Master mix (Roche). Target gene expression was normalized to *UBC* levels[118]. Three independent experiments were performed. Statistical significance was computed using either a Student's *t* test in Microsoft Excel (Microsoft 365 16.61) or ANOVA with post-hoc Tukey's test with an online tool (https://astatsa.com/OneWay_Anova_with_TukeyHSD/). Graphs were generated in R (2021.09.1 Build 372) using the ggplot2 package (3.3.5). Primers used for gene expression analysis are listed in Supplementary Data 12.

For RNA-seq, total RNA was extracted following the method described above. Sequencing libraries were generated based on the Brad-seq method[119]. Briefly, mRNA was fragmented by heating and with magnesium and annealed to an adapter-containing oligonucleotide for cDNA synthesis. After cDNA synthesis by RevertAid RT enzyme (Thermo Fisher Scientific), cDNA was purified and size-selected with Agencourt AMPure XP beads (Beckman). Then, 5′ adapter addition, PCR enrichment, and index-sequence addition were conducted to generate the sequencing libraries. Agarose gel electrophoresis was conducted to confirm DNA fragment sizes. Sequencing was carried out as 50-bp single-end reads on a Next-Seq 500 instrument (Illumina). The resulting bcl files were converted into fastq files by bcl2fastq (Illumina). Mapping of sequences to the Arabidopsis TAIR10 reference genome was performed using BOWTIE2.0 (version 1.2.2) with the options "--all -- best --strata --trim5 8". The number of reads for each gene was then counted, and the false discovery rate (FDR), Log concentration (Conc), and Log fold-change were obtained using the edgeR package[120]. To identify differentially expressed genes, an FDR with an adjusted $p < 0.05$ was used. The sequencing data were deposited into the DNA Data Bank of Japan (DRA014836: wild type vs. *dad1*, wild type vs. *snac* septuple mutant).

## Cis-element identification using RNA-seq data
To identify *cis*-motif enrichment in 70 high-confidence DAD1 downstream genes, 1000 bp of upstream Col-0 sequence from the TAIR10 genome was downloaded from the TAIR website (https://www.arabidopsis.org) for each of the 70 genes. Conserved motifs were discovered with the Multiple Em Motif Elucidation (MEME) program (MEME Suite 5.4.1) (http://meme-suite.org/tools/meme)[121]. After motif identification, each motif was compared against the JASPAR database (http://jaspar.genereg.net) using Tomtom (http://meme-suite.org/doc/tomtom.html?man_type=web)[122,123]. Sequence logos were downloaded from the Tomtom website in jpg format.

## Conserved promoter and *cis*-element identification
To identify conserved promoter sequences and *cis*-motifs, the *ANAC102* promoter sequences from nine *Brassica* species were downloaded from Gene Slider as txt files (http://bar.utoronto.ca/geneslider/)[67]. To determine the extent of conservation of the *ANAC102* promoter across these nine species, phylogenetic shadowing was conducted[67–71]. Text files were converted into fasta files by Textedit for Mac (version 1.7) and uploaded to the mVISTA website (https://genome.lbl.gov/vista/mvista/submit.shtml). Images of conservation bit scores and alignment of the *ANAC102* promoter were obtained directly from the Gene Slider website. G-box sequences near MYC2-binding peaks were obtained from Gene Slider. Using the alignment downloaded from the website, alignments and sequence logos were created using Clustal Omega (https://www.ebi.ac.uk/Tools/msa/clustalo/) and WEBLOGO (https://weblogo.berkeley.edu) in jpg format, respectively.

## ChIP−qPCR
The ChIP assay was conducted according to a previously described protocol with minor modifications[124]. For ChIP−qPCR, 600 mg of inflorescences were harvested. Inflorescences were fixed with 1% (v/v) formaldehyde in PBS for 15 min under vacuum, followed by an incubation in 125 mM glycine for 5 min to stop the fixation reaction, and frozen in liquid nitrogen. The frozen inflorescences were homogenized with a mortar and pestle and dissolved into NEB buffer (100 mM MOPs, 10 mM $MgCl_2$, 0.25 M sucrose, 5% [w/v] dextran T-40, 2.5% [w/v] Ficoll 400, 40 mM β-mercaptoethanol, and 25 μl protease inhibitor per sample) to obtain DNA−protein complexes. The solution was filtered through two layers of Miracloth (Merck) and centrifuged. After discarding the supernatant, the resulting green pellet was resuspended in nuclei lysis buffer (50 mM Tris-HCl, 10 mM EDTA and 1% [w/v] SDS) and kept on ice for 30 min with occasional tapping. Volume up with ChIP dilution buffer was conducted, followed by sonication (TOMY UD201) five times on ice. After removal of debris by centrifugation, the DNA−protein complexes were immunoprecipitated with specific antibodies and Protein A magnetic beads (Thermo Fisher Scientific). Anti-GFP antibody (SAB4301138, Sigma), anti-MYC antibody (9B11, Cell Signaling), anti-FLAG M2 antibody (F3165, Sigma-Aldrich), anti-H3K9ac antibody (31-1054-00, ReMAb Bioscience), and anti-Pol II antibody (ab817, Abcam) were used. The beads were washed with low-salt buffer (20 mM Tris-HCl, 0.1% [w/v] SDS, 1% [v/v] Triton X-100, 2 mM EDTA and 500 mM NaCl), LiCl buffer (0.25 M LiCl, 10 mM Tris-HCl, 1% [v/v] IGEPAL-CA630, 1% [w/v] deoxycholate and 1 mM EDTA), and 0.5× TE buffer (10 mM Tris-HCl and 1 mM EDTA), twice each. The DNA−protein complexes were then eluted from the beads in nuclei-lysis buffer at 65 °C. To reverse the crosslinking, NaCl was added to the eluates and incubation was conducted at 65 °C overnight. Next, DNA was purified using a Qiaquick DNA Purification Kit (Qiagen). The resulting ChIP DNA was subjected to quantitative PCR (qPCR) with a Light Cycler 480 (Roche) instrument and Light Cycler 480 release 1.5.1.62 SP software (Roche) with FastStart DNA Essential DNA Green Master (Roche) under the following conditions: 95 °C for 5 min, 55 cycles at 95 °C for 10 s, 60 °C for 10 s, and 72 °C for 10 s. Primer sequences are listed in Supplementary Data 12.

## ChIP-seq

The ChIP-seq assay was conducted as described with minor modifications[125]. For ChIP-seq, 2 g of inflorescences were harvested and immediately frozen in liquid nitrogen and stored until use. The tissues were ground to a fine powder with an ice-cold mortar and pestle and fixed in nuclei-isolation buffer (10 mM HEPES, 1 M sucrose, 5 mM KCl, 5 mM MgCl₂ and 5 mM EDTA) for 10 min. Subsequently, the fixation reaction was quenched by adding glycine at a final concentration of 125 mM. The resulting solution was filtered through two layers of Miracloth (Merck) to remove debris and centrifuged to obtain a white pellet. The resulting pellet was resuspended in nuclei-isolation buffer, overlayed onto nuclei separation buffer (10 mM HEPES, 1 M sucrose, 5 mM KCl, 5 mM MgCl₂, 5 mM EDTA and 15% [w/v] Percoll), and centrifuged. The chromatin pellet was dissolved into SDS lysis buffer (50 mM Tris-HCl, 1% [w/v] SDS and 10 mM EDTA), further mixed with ChIP dilution buffer (50 mM Tris-HCl, 1% [w/v] SDS, 10 mM EDTA and 10% [w/v] sodium deoxycholate), and sonicated with a Covaris S2 sonicator (Covaris) to produce DNA fragments of varying sizes ranging from 200 to 400 bp in length. Immunoprecipitation was performed using an anti-GFP antibody (SAB4301138, Sigma) and Dynabeads protein A (Thermo Fisher Scientific). The beads were washed with low-salt RIPA buffer (50 mM Tris-HCl, 150 mM NaCl, 1 mM EDTA, 0.1% [w/v] SDS, 1% [v/v] Triton X100 and 0.1% [w/v] sodium deoxycholate), high-salt RIPA buffer (50 mM Tris-HCl, 600 mM NaCl, 1 mM EDTA, 0.1% [w/v] SDS, 1% [v/v] Triton X100 and 0.1% [w/v] sodium deoxycholate), LNDET (10 mM Tris-HCl, 250 mM LiCl, 1% [v/v] IGEPAL CA-630, 1% [w/v] sodium deoxycholate and 1 mM EDTA), and TE buffer (10 mM Tris-HCl and 1 mM EDTA). Next, chromatin was eluted in ChIP direct-elution buffer (10 mM Tris-HCl, 300 mM NaCl, 5 mM EDTA and 0.5% [w/v] SDS) at 65 °C overnight. After digestion of RNA and protein by RNase (Qiagen) and Proteinase K (Roche), ChIP DNA was isolated using a Monarch PCR and DNA cleanup Kit (Monarch). Libraries were prepared using a ThruPLEX DNA-seq Kit (Rubicon Genomics) according to the manufacturer's instructions. Dual size selection was performed using Agencourt AMPure XP beads (Beckman Coulter). The libraries were pooled and sequenced on a Next-Seq 500 instrument (Illumina). Two independent biological replicates were analyzed. The sequence data were deposited into the DNA Data Bank of Japan (DRA014637: gANAC102-GFP ChIP).

Mapping was conducted on the NIG supercomputer at the TOIS National Institute of Genetics. After sequencing, fastq files were quality-filtered. The trimmed reads were mapped to the Arabidopsis TAIR10 reference genome using BOWTIE2.0 (version 1.2.2) to generate sam files[126]. Using SAMtools (version 1.10), sam files were converted into bam files[127]. To visualize multiple peaks, the Integrative Genomics Viewer (IGV; version 2.4.14) was used[128].

## Analysis of publicly available binding and epigenome datasets

Publicly available sequencing datasets were used to visualize chromatin states at genes of interest. MYC2 and MED25 binding (CRA001078), H3K9ac modification (GSE67776, GSE67777, and GSE67778), nucleosome occupancy (GSE50242), and DNase I-hypersensitive site (GSE34318) data were obtained from the National Genomics Data Center (https://bigd.big.ac.cn) or Gene Expression Omnibus (https://www.ncbi.nlm.nih.gov/geo/)[57–59,129]. For MYC2 and MED25 peaks, downloaded fastq files were mapped as described above. For H3K9Ac, nucleosome occupancy, and DNase I hypersensitive assay, bedGraph, Wiggle, and bed files were downloaded from the databases and used. Peaks were visualized in IGV (version 2.4.14)[128].

## Formaldehyde-assisted isolation of regulatory elements (FAIRE)

To determine the sequences of genomic regions associated with regulatory activity, FAIRE–qPCR was conducted[130]. For FAIRE–qPCR, 600 mg of inflorescences were used. Inflorescences were fixed in 1% (v/v) formaldehyde in PBS or PBS only for 15 min under vacuum,

incubated in 0.125 M glycine for 5 min to stop the fixation reaction, and frozen in liquid nitrogen. The frozen inflorescences were homogenized into a fine powder with a mortar and pestle to obtain DNA–protein complexes. The resulting powder was resuspended into Buffer 1 (10 mM Tris-HCl, 400 mM sucrose, 5 mM β-mercaptoethanol, 0.1 mM PMSF, and 1 tablet of Complete protease inhibitor cocktail per 50 mL Buffer 1) and incubated for 10 min at room temperature. After filtering the homogenized suspension through four layers of Miracloth (Merck), the chromatin was centrifuged. The resulting white pellet was resuspended in Buffer 2 (10 mM Tris-HCl, 250 mM sucrose, 10 mM MgCl₂, 1% [v/v] Triton X100, 5 mM β-mercaptoethanol, 0.1 mM PMSF and 0.5 tablet of Complete protease inhibitor cocktail per 10 mL Buffer 2). Centrifugation, discarding of the supernatant, and resuspension were repeated twice. The final pellet was dissolved into ice-cold Buffer 3 (10 mM Tris-HCl, 1.7 M sucrose, 2 mM MgCl₂, 0.15% [v/v] Triton X100, 5 mM β-mercaptoethanol, 0.1 mM PMSF and 0.5 tablet of Complete protease inhibitor cocktail per 10 mL Buffer 3). After centrifugation, the supernatant was discarded and the chromatin pellet was resuspended into ice-cold nuclei-lysis buffer (50 mM Tris-HCl, 10 mM EDTA, 1% [w/v] SDS, 0.1 mM PMSF and 1/4 of a Complete protease inhibitor cocktail per 5 mL Buffer nuclei-lysis buffer). The solution was sonicated (TOMY UD201) five times. After removal of debris by centrifugation, nucleosome-depleted regions were isolated by phenol–chloroform-based DNA extraction. Next, DNA was further purified using a Qiaquick DNA Purification Kit (Qiagen). The resulting ChIP DNA was subjected to qPCR with a Light Cycler 480 (Roche) instrument and Light Cycler 480 release 1.5.1.62 SP software (Roche). qPCR was conducted with FastStart DNA Essential DNA Green Master (Roche) under the following conditions: 95 °C for 5 min, 55 cycles at 95 °C for 10 s, 60 °C for 10 s, and 72 °C for 10 s. Primer sequences are listed in Supplementary Data 12.

## In situ proximity ligation assay (PLA)

To observe protein–protein interactions in petal cells, in situ PLA was performed using a Duolink In Situ Red Starter Kit Mouse/Rabbit (Sigma) according to the manufacturer's instructions with minor modifications[66]. Petals at position 3 were harvested and fixed in PBS with 4% (w/v) paraformaldehyde for 40 min. After washing with PBS, tissues were roughly chopped with a razor blade on a glass slide with lysis buffer (15 mM Tris-HCl, 2 mM EDTA, 0.5 mM spermine, 80 mM KCl, 20 mM NaCl and 0.1% [v/v] Triton X100). The resulting solution containing DNA was mixed with suspension buffer (100 mM Tris-HCl, 50 mM KCl, 2 mM MgCl₂, 5% [w/v] sucrose and 0.05% [v/v] Tween 20) and filtered through two layers of Miracloth (Merck) to remove large pieces of plant tissues. A drop of nuclei solution was then put onto a glass slide and dried overnight in darkness at room temperature. A drop of Blocking solution (Sigma) was added to the nuclei sample area. The slides were incubated at 37 °C for 30 min. After removal of the Blocking solution, the primary antibody diluted in Antibody Diluent (Sigma) was added. To prepare the primary-antibody solutions, rabbit anti-GFP (ab290, Abcam), rabbit anti-HA (12CA5, Roche), and mouse anti-FLAG (F3165, Sigma-Aldrich) were diluted 1:100. Slides with nuclei samples were incubated at 4 °C overnight and washed in a wash bottle with Wash Buffer A (DUO82046, Sigma) for 5 min twice. After addition of PLA probe solution (5× PLA probe Anti-Mouse PLUS:5× PLA probe Anti-Rabbit MINUS:1× Antibody Diluent = 1:1:8, v/v/v), incubation was conducted at 37 °C for 1 h. Slides were then washed in a bottle with Wash Buffer A (DUO82046, Sigma) for 5 min twice. Subsequently, Ligation–Ligase solution (Sigma) was dropped onto nuclei samples and incubation was carried out at 37 °C for 1 h. After washing the Ligation–Ligase solution away with Buffer A, Amplification–Polymerase solution (Sigma) was added to nuclei samples on the glass slides, and kept at 37 °C for 100 min. The slides were washed in Wash Buffer B (DUO82048, Sigma). Samples were

mounted in Duolink in situ DAPI (Sigma) solution and observed on an LSM700 (ZEISS) confocal laser scanning microscope. PLA signals were counted as reported previously[66]. Graphs were generated in either R (2021.09.1 Build 372) using the ggplot2 package (3.3.5) or in Microsoft Excel (Microsoft 365 16.61). Student's t-test was performed in Microsoft Excel (Microsoft 365 16.61).

### Immunoblot analysis

For immunoblot analysis, protein extracts were prepared from petals at the indicated position in 100 μL of SDS sample buffer. All of the eluates were loaded onto 4–12% SDS–PAGE gel (Invitrogen) and separated in a Blot Mini Gel tank (Thermo Fisher Scientific). The resulting gel was subjected to immunoblotting using an iBlot2 dry-blotting system (Thermo Fisher Scientific). Rabbit polyclonal anti-GFP (ab290, Abcam, 1:1,000 diluted), anti-histone H3 (ab1791, Abcam, 1:1,000 diluted), and anti-rabbit IgG HRP conjugate (1:5,000 diluted, Thermo Fisher Scientific) were used as primary and secondary antibodies. Signals were detected with chemiluminescence HRP substrates (Millipore) and Image analyzer (LAS4000, GE healthcare). Molecular mass was determined by MagicMark XP (Thermo Fisher Scientific). After detection of signals by LAS4000, each membrane was dried, washed with PBS, and stained with Coomassie brilliant blue at room temperature for 30 min (CBB: Thermo Fisher Scientific). The resulting membrane was scanned using an ApeosPort C4570 (FUJIFILM). Histone H3 or CBB staining was used as a loading control. Signal intensity of each band was quantified in ImageJ (NIH). Statistical significance was computed using Student's t test in Microsoft Excel (Microsoft 365 16.61).

### Reporting summary

Further information on research design is available in the Nature Portfolio Reporting Summary linked to this article.

## Data availability

ChIP-seq data were collected from the National Genomics Data Center or Gene Expression Omnibus (MYC2/MED: CRA001078; H3K9ac: GSE67776, GSE67777, and GSE67778; nucleosome occupancy: GSE50242; DNase I-hypersensitive site: GSE34318).The RNA-seq and ChIP-seq data have been deposited in the DDBJ database (DRA014836, DRA014637). Flower-specific expression data was obtained from the TraVA website (http://travadb.org/browse/). All databases associated with the software and packages used in the study are described in the "Methods" section. Source data are provided with this paper.

## Code availability

Main source codes that are used in this work are available on the GitHub repository [https://zefeng2018.github.io/Plant-Gene-Expression-Prediction/]. Please refer to "Methods" above for details of key parameters and package versions.

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

## Acknowledgements

We thank Akie Takahashi, Hiroko Egashira, Mizuki Shinoda, Rie Shimano, Kyoko Sunuma, and Mikiko Higashiura for technical support for molecular genetics; Ayumi Furuta for technical support for RNA-seq (Chubu University); Dr. Mayuko Sato (RIKEN CSRS) for technical support for electron microscopy, and Sachi Ando for checking a draft of this manuscript. We also thank Dr. Moritz Nowack for the *BFN1pro:nGFP* construct, Dr. Kazuko Shinozaki for the *snac* mutant, Dr. Hao Yu for the *jin1-8 MYC2pro:MYC2–FLAG* seeds, Dr. Sachihiro Matsunaga for the *HDA6pro:HDA6–GFP* construct, Dr. Doris Wagner for the *med25-4, HDA19pro:HDA19–GFP, 35Spro:MED25–GFP*, and *35Spro:TPL–GFP* seed and constructs, Dr. Jiaojiao Wang for the *myc2 myc3 myc4* triple mutant seeds, and Dr. Kohki Yoshimoto for the *35Spro:GFP–ATG8a* construct and *atg5-1* seeds. Seeds for *dad1-3, aos, coi1-51*, and *35Spro:JAZ9–VENUS* were obtained from TAIR. Computations were partially performed on the NIG supercomputer at the ROIS National Institute of Genetics. This work was supported by a grant from a JSPS KAKENHI Grant-in-Aid for Scientific Research B (No. 18H02465, 23H02503), a Grant-in-Aid for challenging Exploratory Research (No. 19K22431), a Grant-in-Aid for Transformative Research Area (A) (No. 21H05663, 23H04968), a grant from the Daiichi Sankyo Foundation of Life Science, a grant from the LOTTE Foundation, and a grant from Ohsumi Frontier Science Foundation to N.Y., a Sasagawa Scientific Research Grant from Sasagawa Foundation to A.U., a JSPS KAKENHI Research Fellowship for Young Scientists (19J00658) to T.H., a JSPS KAKENHI Grant-in-Aid for Scientific Research on Innovative Areas (22H04723), the Takeda Science Foundation, and the Kato Memorial Bioscience Foundation, Japan Science and Technology Agency 'Precursory Research for Embryonic Science and Technology' (JPMJPR22D3) to M.S., Cabinet Office, Government of Japan, Cross-ministerial Strategic Innovation Promotion Program (SIP), 'Technologies for Smart Bio-industry and Agriculture' to Y.I., a JSPS KAKENHI Grant-in-Aid for Scientific Research C (No. 15H05955) to T.S., a Grant-in-Aid for Scientific Research on Innovative Areas (No. 22H04649) to T.G., a JSPS KAKENHI Grant-in-Aid for Scientific Research on Innovative Areas (No. 17H06475) to K.T. and a JSPS KAKENHI Grant-in-Aid for Scientific Research on Innovative Areas (No. 21K19266), a JSPS KAKENHI Grant-in-Aid for Scientific Research A (No. 20H00470), and a Grant-in-Aid for Transformative Research Area (A) (No. 22H05176) to T.I.

## Author contributions

Conceptualization: N.Y. (lead) and all other authors (supporting); data curation: N.Y.; formal analysis: Y.F., H.Y., T.H., A.U., H.I., N.K., N.T., K.K., Y.I., T.S., K.T. and N.Y.; funding acquisition: H.Y., A.U., M.S., Y.I., T.S., T.G., K.T., T.I. and N.Y.; investigation: Y.F., H.Y., T.H., A.U. and N.Y. (lead) and H.I., N.K., N.T., K.K., M.S., S.I., Y.I., T.S., T.G. and K.T. (supporting); project administration: N.Y.; software: N.Y. (lead) and T.S. (supporting); supervision: T.I. and N.Y.; validation: T.I. and N.Y.; visualization: N.Y.; writing: T.H., M.A.P. and N.Y. (original draft) and all authors (review and editing).

## Competing interests

The authors declare no competing interests.

## Additional information

[1]Division of Biological Science, Graduate School of Science and Technology, Nara Institute of Science and Technology, 8916-5, Takayama, Ikoma, Nara 630-0192, Japan. [2]Smurfit Institute of Genetics, Trinity College Dublin, D02 PN40 Dublin, Ireland. [3]Kazusa DNA Research Institute, Kisarazu, Chiba 292-0818, Japan. [4]RIKEN Center for Sustainable Resource Science, 1-7-22 Suehiro-cho, Tsurumi-ku, Yokohama 230-0045, Japan. [5]RIKEN BioResource Research Center, 3-1-1 Koyadai, Tsukuba, Ibaraki 305-0074, Japan. [6]Precursory Research for Embryonic Science and Technology, Japan Science and Technology Agency, Kawaguchi-shi, Japan. [7]Graduate School of Bioagricultural Sciences, Nagoya University, Furo-cho, Chikusa-ku, Nagoya, Aichi 464-8601, Japan. [8]Department of Biological Chemistry, College of Bioscience and Biotechnology, Chubu University, 1200 Matsumoto-cho, Kasugai, Aichi 487-8501, Japan. [9]These authors contributed equally: Yuki Furuta, Haruka Yamamoto. ✉e-mail: itot@bs.naist.jp; nobuy@bs.naist.jp

