## [Peer Review File · Nature Communications]

Petal abscission is promoted by jasmonic acid-induced autophagy at Arabidopsis petal basesREVIEWER COMMENTS

Reviewer #1 (Remarks to the Author):

In this manuscript, Yamamoto et al. reported that petal abscission is regulated in a jasmonic acid (JA)-dependent spatiotemporal manner, through directing the local cell fate determination via autophagy at the base of petals. The authors found that JA biosynthesis mutants showed delayed petal abscission compared with WT. And during petal abscission, JA is accumulated at the base of the petals and triggers cell differentiation, and ROS levels are also regulated by JA. Besides, the authors performed omics and found that ANAC102, one of the targets of the JA signaling pathway, controls petal abscission. The authors examined the spatiotemporal expression of ANAC102 during petal abscission. And they found that ANAC102 triggers local ROS accumulation and cell death via AUTOPHAGY-RELATED GENES induction, and autophagy at the petal base causes maturation. The authors presented substantial data, and revealed some interesting findings. And most of the conclusions are supported by data presented. And I have the following suggestions.

1. In figure 1a, the authors defined the stages of petal abscission from e.g. -3 to 3. And in figure 1b, they showed the petal abscission phenotypes in different genotypes, mainly JA related. I suggest that the authors show zoom-in phenotypes at key stages in different genotypes (preferably under high magnification stereomicroscope), to help the readers better visualize the petal abscission phenotypes.
2. Line 173, 'To gain more insight into petal abscission, we sectioned the petal base from position 3 flowers...' . The authors found that 'At position d, WT petal cells were partly filled with large vacuoles, indicating cell differentiation (Fig. 1j; top). Notably, we only noticed partial vacuolation of ag and dad1 cells at position d. Furthermore, autophagic body-like structures did not accumulate in the vacuoles of WT petal base cells, while those of ag and dad1 contained several autophagic body-like structures in their vacuoles (Fig. 1k, l).' They concluded that 'JA promote petal abscission through petal cell differentiation at their base'. I wonder whether the cellular position and cell size of the petal base at position 5/6/7/8 in the dad1 mutant are similar compared to WT at position 3? Whether JA-treated dad1, aos mutants showed recovered phenotypes at position d, similar compared to WT? And also, whether the coi1, myc2myc3myc4, anac102 mutants show the similar phenotype?
3. Related to the second question, whether the reduced DAB staining and trypan blue staining in different JA-related mutants compared with WT at petal base position 3, is due to delayed ROS accumulation in the JA-related, i.e., normal staining at position 5 or 6 or 7 or 8 in JA mutants, compared with WT position 3; or on the other hand, ROS accumulation is reduced at whatever petal positions in the JA mutants?
4. Line 204, the authors 'detected DAD1 promoter activity (pDAD1::GUS) and DAD1 protein accumulation (pDAD1::DAD1-GFP) at the same stages during stamen development (Fig.2f, g)... DAD1-GFP did not accumulate in petals (Fig. 2h).' They discussed that '...JA production in the stamens for proper pollen maturation and anther dehiscence and coordination of petal development and flower opening by diffusion or active transport to petals.' I am curious whether other JA biosynthesis genes are also expressed in stamens, but not expressed at the base of petals? In addition, whether the stamen abscission and petal abscission are correlated? And whether JA is also involved in stamen abscission?

Reviewer #2 (Remarks to the Author):

In this manuscript, the authors investigated the JA-induced transcriptional changes that lead to petal abscission in Arabidopsis. I am not an expert on hormones or transcriptional responses, so I will focus on the cell biology part in my report. In the current state, the manuscript is not conclusively linking autophagy to petal abscission. Please see below my detailed comments:

1/ Title-Jasmonic acid-induced instead of mediated to avoid having 2 mediated in the title?

2/ Fig1k/l-autophagic body-like structures is a very vague term that is not backed up by any data. There are various vacuolar trafficking pathways that could be the source of those vesicles. The authors need to at least back this up with some ATG8 immunogold labeling to be able to say that these are autophagic vacuoles.

3/ My main concern for the autophagy part is the lack of causal evidence directly linking JA-induced transcriptional responses to autophagy. The author's data and previous studies have shown that JA could induce autophagy, but how this autophagy mediates petal abscission is not clear not us. In addition, the authors are well aware of the pleiotropic phenotypes associated with atg mutants. Stress hormones and other stress responses are all induced in these mutants since they are constantly lacking a core homeostatic pathway. To directly link abscission with autophagy, the authors need to use a tissue-specific CRISPR knockout strategy and use petal abscission-specific promoters. Without this evidence, all the other evidence will be circumstantial.

4/ Fig8 ideally should be performed in a +/-Concanamycin A setup, so that we can see the autophagic flux.

Reviewer #3 (Remarks to the Author):

This study tells the story of the mechanism of JA / NAC module in Arabidopsis petal abscission by regulating ROS and autophagy in the petal base, and this article reveals the role of proper autophagy regulated by JA in organ abscission. The research evidence is detailed and accurate, and I have some points to communicate with the author.

1.The significance annotation in the chart confused me. For example, in Figure 1c, a, b, c are shown, but an asterisk is marked above the bar chart, and the standard should be unified.

2.How do the authors understand that treating ag and dad1 mutants with JA does not fully complement their petal shedding phenotype?

3.Whether genes downstream of ANAC102 regulates ROS production in the results of RNA-seq and Chip-seq.

4.Figure 7 needs to complement the abscission phenotypes of atg8a, the staining results of DAB and Trypan blue, and the abscission phenotype of atg7 can be included in the supplementary data.

Extended Data Fig. 3 should be annotated with position.

Point-by-point responses to reviewers' comments

We wish to thank the reviewers very much for their thoughtful and helpful comments that have resulted in the manuscript being strengthened. Please find below our responses to each point.

Best wishes,

Nobutoshi Yamaguchi, on behalf of all the authors.

Responses to Reviewer #1

General comment by Reviewer #1

In this manuscript, Yamamoto et al. reported that petal abscission is regulated in a jasmonic acid (JA)-dependent spatiotemporal manner, through directing the local cell fate determination via autophagy at the base of petals. The authors found that JA biosynthesis mutants showed delayed petal abscission compared with WT. And during petal abscission, JA is accumulated at the base of the petals and triggers cell differentiation, and ROS levels are also regulated by JA. Besides, the authors performed omics and found that ANAC102, one of the targets of the JA signaling pathway, controls petal abscission. The authors examined the spatiotemporal expression of ANAC102 during petal abscission. And they found that ANAC102 triggers local ROS accumulation and cell death via AUTOPHAGY-RELATED GENES induction, and autophagy at the petal base causes maturation. The authors presented substantial data, and revealed some interesting findings. And most of the conclusions are supported by data presented. And I have the following suggestions.

General Response

We appreciate the constructive feedback provided by Reviewer 1. As indicated in the following **Responses**, we have carefully considered all of your comments and suggestions during the revision of our manuscript. In addition, we have included 16 new supplementary figures.

The major changes can be summarized as:

1. Observation of JA and *atg* mutants under a high-magnification stereomicroscope.
2. Temporal observation of petal-cell dynamics in JA-related mutants via SEM.

3. Temporal observation of ROS accumulation in JA-related mutants using DAB staining.
4. Stamen-abscission measurements conducted on selected JA-related mutants.
5. Expression analysis of JA biosynthetic genes.

We believe that the suggested experiments have provided more precise support for our hypothesis compared to the original version. Please refer to our point-by-point responses below.

Comment 1 by Reviewer #1

In figure 1a, the authors defined the stages of petal abscission from e.g. -3 to 3. And in figure 1b, they showed the petal abscission phenotypes in different genotypes, mainly JA related. I suggest that the authors show zoom-in phenotypes at key stages in different genotypes (preferably under high magnification stereomicroscope), to help the readers better visualize the petal abscission phenotypes.

Response

Based on this suggestion, we have added magnified images of inflorescences to the revised version of our supplementary figures. To show the progression of petal abscission in both WT and JA-related mutants, as well as in *anac* and *atg* mutants, we examined petal abscission in floral-bud clusters and in petals at positions +3, +5, +7, and +9 (Extended Data Figures 1, 13, 22, 32, 33). WT petals were typically detached by position +5, followed by subsequent fruit elongation. Unlike the WT, most of the mutants examined in this study still retained their petals at position +5. We believe that these five new figures will help the readers better visualize the progression of petal abscission.

Comment 2 by Reviewer #1

*Line 173, 'To gain more insight into petal abscission, we sectioned the petal base from position 3 flowers...' . The authors found that 'At position d, WT petal cells were partly filled with large vacuoles, indicating cell differentiation (Fig. 1j; top). Notably, we only noticed partial vacuolation of *ag* and *dad1* cells at position d. Furthermore, autophagic body-like structures did not accumulate in the vacuoles of WT petal base cells, while those of *ag* and *dad1* contained several autophagic body-like structures in their vacuoles (Fig. 1k, l).' They concluded that 'JA promote petal abscission through petal cell differentiation at their base'. I wonder whether*

the cellular position and cell size of the petal base at position 5/6/7/8 in the dad1 mutant are similar compared to WT at position 3? Whether JA-treated dad1, aos mutants showed recovered phenotypes at position d, similar compared to WT? And also, whether the coi1, myc2myc3myc4, anac102 mutants show the similar phenotype?

Response

Indeed, we should have examined petal cell differentiation not only at earlier stages, but also at later stages. We have now conducted all three suggested experiments to better explore temporal changes in cell vacuolation, the effects of JA on cell vacuolation in the respective mutants, and the phenotypes of other JA-related mutants using SEM.

1. We compared petal cells at position +3 in WT with those at position +7 in *dad1*. We observed that petal cells in *dad1* become vacuolated at position +7 (Extended Data Figure 6).

2. We performed SEM using JA-treated WT and *dad1* petals to observe the rescue of the mutant phenotype. In accordance with the timing of petal abscission, the partial-vacuolation phenotype observed in *dad1* petal cells was rescued after JA treatment (Extended Data Figure 7). Thus, petal cells in JA-treated *dad1* at position +3 were filled with large vacuoles, like the WT.

3. Finally, we also observed cellular changes in *coi1* and *myc2 myc3 myc4* petals (Extended Data Figure 6). Similar to what was observed in *ag* and *dad1* cells, we observed partial vacuolation in the *coi1* and *myc2 myc3 myc4* mutants.

These suggested experiments support the original hypothesis that JA promotes petal abscission by affecting petal-cell differentiation at the petal base. We thank you for these suggestions.

Comment 3 by Reviewer #1

3. Related to the second question, whether the reduced DAB staining and trypan blue staining in different JA-related mutants compared with WT at petal base position 3, is due to delayed ROS accumulation in the JA-related, i.e., normal staining at position 5 or 6 or 7 or 8 in JA mutants, compared with WT position 3; or on the other hand, ROS accumulation is reduced at whatever petal positions in the JA mutants?

Response

We fully agree with this point. In the revised manuscript, we have included images of DAB staining at two different floral positions: position +3 and +7 (Extended

Data Figures 5, 14, 23). At position +3, just prior to petal abscission, the WT exhibited significantly higher ROS accumulation at its petal base compared to the JA-related mutants. However, at position +7, ROS accumulation in the JA-related mutants was similar to that at position +3 of the WT. These findings suggest that ROS accumulation in JA-related mutants is closely linked to the *timing* of petal abscission rather than to a general deficiency in ROS production. The time-course analysis of ROS accumulation in both WT and *dad1* further supports this interpretation (Extended Data Figure 4).

Comment 4 by Reviewer #1

4. Line 204, the authors 'detected *DAD1* promoter activity (*pDAD1::GUS*) and *DAD1* protein accumulation (*pDAD1::DAD1-GFP*) at the same stages during stamen development (Fig.2f, g)... *DAD1-GFP* did not accumulate in petals (Fig. 2h).' They discussed that '...JA production in the stamens for proper pollen maturation and anther dehiscence and coordination of petal development and flower opening by diffusion or active transport to petals.' I am curious whether other JA biosynthesis genes are also expressed in stamens, but not expressed at the base of petals? In addition, whether the stamen abscission and petal abscission are correlated? And whether JA is also involved in stamen abscission?

Response

Based on this comment, we examined the spatiotemporal expression of the major JA biosynthesis genes *AOS* (Laudert and Weiler, 1998) and *OPR3* (Sander et al., 2000) (Extended Data Figure 9). *AOS* and *OPR3* encode a cytochrome P450 and a 12-oxo-phytodienoic acid (OPDA) reductase, respectively, both of which are required for proper JA biosynthesis. As observed for *DAD1* expression, *AOS* and *OPR3* were specifically expressed in stamen filaments, but not in petals. Therefore, not only is the JA intermediate generated in stamens (via *DAD1*), but the resulting metabolites might also be used to produce JA in stamens via a process catalyzed by *AOS* and *OPR3*.

We also observed the timing of petal and stamen abscission in WT and *dad1* plants and their correlation (Extended Data Figure 2). In the WT, the timing of petal and stamen abscission were well correlated. In the JA-related mutants, the correlation coefficient between the timing of petal and stamen abscission was close to 1.0, suggesting that JA is involved in stamen abscission as well.

Although GUS expression and stamen-abscission data included in this revision provide additional support for our original hypothesis, the mechanisms of JA diffusion or transport between floral organs is not fully addressed yet. In the revised version, we have discussed future prospects for JA in organ abscission (Page 14 line 604-610).

Laudert D, Weiler EW (1998). Allene oxide synthase: a major control point in *Arabidopsis thaliana* octadecanoid signalling. *Plant J.* 15: 675-684.

Sanders PM, Lee PY, Biesgen C, Boone JD, Beals TP, Weiler EW, Goldberg RB (2000). The *Arabidopsis DELAYED DEHISCENCE1* gene encodes an enzyme in the jasmonic acid synthesis pathway. *Plant Cell.* 2000 12: 1041-1061.

Responses to Reviewer #2

General comment by Reviewer #2

In this manuscript, the authors investigated the JA-induced transcriptional changes that lead to petal abscission in Arabidopsis. I am not an expert on hormones or transcriptional responses, so I will focus on the cell biology part in my report. In the current state, the manuscript is not conclusively linking autophagy to petal abscission.

Please see below my detailed comments:

General response

We are grateful to Reviewer #2 for the critical comments, which have helped us improve our paper. As indicated below, we have taken all these comments and suggestions into account in the revised manuscript.

Importantly, we obtained causal evidence to directly link JA-induced transcriptional responses to autophagy, based on tissue-specific knockdown and complementation lines, as suggested. We believe that phenotyping of these two types of transgenic lines has strengthened our results. Furthermore, we performed concanamycin A treatment and observed changes in response to inhibited autophagic flux, similar to those of defects in jasmonic acid. Finally, we carefully rephrased our descriptions, as suggested.

We believe that the suggested experiments provide stronger support for our hypothesis. For details, please refer to our point-by-point responses below.

Comment 1 by Reviewer #2

1. Title-*Jasmonic acid-induced instead of mediated to avoid having 2 mediated in the title?*

Response

Thank you for pointing this out. Based on this suggestion, we have changed the title of our manuscript to the following: 'Petal abscission is promoted by jasmonic acid-induced autophagy at Arabidopsis petal bases'.

Comment 2 by Reviewer #2

2. *Fig1k/l-autophagic body-like structures is a very vague term that is not backed up by any data. There are various vacuolar trafficking pathways that could be the source of those vesicles. The authors need to at least back this up with some ATG8 immunogold labeling to be able to say that these are autophagic vacuoles.*

Response

Thank you for raising this important point. One of our authors, Kiminori Toyooka, successfully performed immunogold labeling using anti-ATG8 antibody in a previous study (Yoshimoto et al., 2014, *Journal of Cell Science*). However, despite his expertise, sectioning of the petal base for immunogold labeling would be technically challenging because the fixation solution is transparent, unlike the osmium solution used for TEM, rendering the flowers invisible during fixation.

To avoid using such an ambiguous term, we rephrased and toned down the descriptions of Fig. 1k and 1l in the revised manuscript. What we had intended to say is that round membrane structures containing cytoplasmic components were observed within vacuoles in the *ag* or *JA*-related mutant backgrounds. In the revised version, we now refer to them as small vesicles within vacuoles (Results, Page 5, lines 189–190) (Discussion, Page 14, lines 589–591). In addition, as indicated by Reviewer 2, we now stress that these vesicles could be generated not only due to defects in autophagy, but also through various vacuolar trafficking pathways (Results, Page 5, lines 190–192). (Discussion, Page 14, lines 589–591).

Yoshimoto K, Shibata M, Kondo M, Oikawa K, Sato M, Toyooka K, Shirasu K, Nishimura M, Ohsumi Y (2014). Organ-specific quality control of plant peroxisomes is mediated by autophagy. *J Cell Sci.* **127**: 1161-1168.

Comment 3 by Reviewer #2

3. My main concern for the autophagy part is the lack of causal evidence directly linking JA-induced transcriptional responses to autophagy. The author's data and previous studies have shown that JA could induce autophagy, but how this autophagy mediates petal abscission is not clear not us. In addition, the authors are well aware of the pleiotropic phenotypes associated with *atg* mutants. Stress hormones and other stress responses are all induced in these mutants since they are constantly lacking a core homeostatic pathway. To directly link abscission with autophagy, the authors need to use a tissue-specific CRISPR knockout strategy and use petal abscission-specific promoters. Without this evidence, all the other evidence will be circumstantial.

Response

We fully agree with this critical point. To provide causal evidence, we have performed two parallel experiments: 1) Tissue-specific knockdown and 2) tissue-specific genetic complementation. In the revised manuscript, these data are presented in the new Figure 8.

1. Tissue-specific knockdown.

As suggested by Reviewer 2, CRISPR–TSKO would be one of the best techniques for efficient mutagenesis in specific cell types, tissues, or organs (Decaestecker et al., 2019, Plant Cell). Although one of our authors, Tatsuaki Goh, has a CRISPR–TSKO gene-knockout tool set based on the Golden Gate system, we have not yet established this gene-knockout technique due to some technical issues. Instead, we generated transgenic Arabidopsis lines expressing an artificial microRNA targeting *ATG5* (*amiR-ATG5*) under the control of the *NAC102* promoter, which drives transcription at the base of position +2 flowers onward in the *atg5 gATG5–GFP* background (Schwab et al., 2006, Plant Cell; Goh et al., 2022, Development). The *amiR* construct leads to lower *ATG5–GFP* fluorescence specifically at the petal base prior to abscission. The *NAC102_{pro}:amiR-ATG5* and *NAC102_{pro}:amiR-ATG7* plants have delayed petal abscission, fewer dead cells, and lower DAB accumulation in petal bases. These results suggest that local autophagy activity at the petal base is required for petal abscission.

2. Tissue-specific rescue.

To further clarify whether activation of autophagy at the petal base is sufficient for petal abscission, we generated plants expressing *mVENUS-ATG7* in the *atg7* mutant background under the control of the *NAC102* promoter, which drives transcription at the base of position +2 flowers onward. We detected dot-like *mVENUS-ATG7* signals at the petal base. Phenotyping, as well as trypan blue and DAB staining, revealed that the delay in petal abscission seen in the *atg7* mutant was restored to WT in these transgenic plants. These results suggest that the activation of autophagy at the petal base is sufficient to trigger petal abscission.

Decaestecker W, Buono RA, Pfeiffer ML, Vangheluwe N, Jourquin J, Karimi M, Van Isterdael G, Beeckman T, Nowack MK, Jacobs TB. CRISPR-TSKO: A Technique for Efficient Mutagenesis in Specific Cell Types, Tissues, or Organs in *Arabidopsis* (2019). *Plant Cell* **31**: 2868-2887.

Schwab R, Ossowski S, Riester M, Warthmann N, Weigel D. (2006) Highly specific gene silencing by artificial microRNAs in *Arabidopsis*. *Plant Cell*. 18: 1121-1133.

Goh T, Sakamoto K, Wang P, Kozono S, Ueno K, Miyashima S, Toyokura K, Fukaki H, Kang BH, Nakajima K. (2022). Autophagy promotes organelle clearance and organized cell separation of living root cap cells in *Arabidopsis thaliana*. *Development* **149**: dev200593.

Comment 4 by Reviewer #2

4. *Fig8 ideally should be performed in a +/- concanamycin A setup, so that we can see the autophagic flux.*

Response

In Fig. 8, we have examined the effect of JA signaling on GFP-ATG8a dynamics during petal abscission. However, how inhibition of autophagy affects GFP-ATG8a dynamics during petal abscission was unclear. To determine whether we would observe similar changes in GFP-ATG8a dynamics in plants with inhibited autophagic flow and in plants with jasmonic acid defects, we have now conducted concanamycin A treatment during petal development (Extended Data Figure 35). As with the JA-related mutants, the degradation of ATG8a was delayed at the petal base upon concanamycin A treatment. This significant delay was confirmed

using immunoblotting-based autophagic-flux assays with concanamycin A-treated petals. These results suggest that the inhibition of autophagy and defects in JA signaling have similar effects on GFP–ATG8a dynamics during petal abscission.

Interestingly, we observed differences in autophagosome formation in petals treated with concanamycin A and in petals from JA-related mutants. Concanamycin A is a specific inhibitor of vacuolar type H⁺-ATPase activity that causes autophagic bodies to accumulate in vacuoles without affecting autophagosome shape. Indeed, we often observed typical ring-shaped autophagosomes in both mock- and concanamycin A-treated petal bases. By contrast, we rarely saw ring-shaped autophagosomes seen in the JA-related mutants, suggesting that JA signaling also affects autophagosome formation either directly or indirectly due to the lack of a core homeostatic pathway. In the revised manuscript, we have included a discussion about both the similarities and differences in the JA- and concanamycin A-mediated pathways.

Responses to Reviewer #3

General comment by Reviewer #3

This study tells the story of the mechanism of JA / NAC module in Arabidopsis petal abscission by regulating ROS and autophagy in the petal base, and this article reveals the role of proper autophagy regulated by JA in organ abscission. The research evidence is detailed and accurate, and I have some points to communicate with the author.

General Response

We appreciate the favorable comments and useful suggestions provided by Reviewer 3. As indicated in the following responses, we have carefully considered all of these comments and suggestions in the revised manuscript.

The major revisions are as follows:

1. We observed the phenotypes of the *atg8a* mutant.
2. We added descriptions of the direct targets of ANAC102 involved in ROS accumulation.
3. We added a detailed explanation of the rescue of defects in *ag* and *dad1* by JA treatment.

We believe that the suggested experiments provided more precise support for our hypothesis compared to the original version. For details, please refer to our point-by-point responses below.

Comment 1 by Reviewer #3

1. The significance annotation in the chart confused me. For example, in Figure 1c, a, b, c are shown, but an asterisk is marked above the bar chart, and the standard should be unified.

Response

We made the requested changes to our figures. We have also expanded and updated the description of the figure labels throughout the manuscript.

Comment 2 by Reviewer #3

*2. How do the authors understand that treating *ag* and *dad1* mutants with JA does not fully complement their petal shedding phenotype?*

Response

We conducted JA treatment via a single spray application with methyl jasmonate. This regimen rescued petal abscission without causing visible secondary effects (e.g. cellular damage), but petal abscission in only 2 or 3 flowers per plant were fully rescued 2 days after application. The remaining flowers were either rescued only partially or not at all. Previously, one of our authors, Sumie Ishiguro, also demonstrated the rescue of a limited number of flowers in JA biosynthesis mutants via a single spray treatment (Ishiguro et al., 2001, *Plant Cell*: Please see Fig. 3E in the *Plant Cell* paper). Therefore, obtaining the proper floral stage before treatment is critical for successful rescue. We believe that the partial rescue was largely due to subtle differences in the stages of flowers before JA treatment.

We believe that the petal-abscission phenotype could be rescued by performing multiple JA treatments. Nevertheless, we are inclined not to pursue this approach because it could potentially trigger a JA-mediated defense response and/or lead to secondary effects and confound the experiment. In the revised manuscript, we have provided a comprehensive description of our methods and the outcomes of the rescue process in the Results section.

Ishiguro S, Kawai-Oda A, Ueda J, Nishida I, Okada K. The *DEFECTIVE IN ANTHWER DEHISCENCE* gene encodes a novel phospholipase A1 catalyzing the

initial step of jasmonic acid biosynthesis, which synchronizes pollen maturation, anther dehiscence, and flower opening in *Arabidopsis* (2001). *Plant Cell* **13**: 2191-2209.

Comment 3 by Reviewer #3

3. Whether genes downstream of ANAC102 regulates ROS production in the results of RNA-seq and Chip-seq.

Response

Indeed, ANAC102 directly controls the expression of genes involved in ROS production based on RNA-seq and ChIP-seq analysis. For example, the *PEROXIDASE 33 (PRX33)* gene encodes a Class III peroxidase that plays a role in generating H₂O₂ (Arnaud et al., 2023, *Plant Physiology*). Another example is the *PEROXIN11d (PEX11d)* gene, which encodes a peroxisomal membrane protein that regulates peroxisome proliferation (Orth et al., 2007, *Plant Cell*). In the revised manuscript, we have added descriptions of a few key genes involved in ROS production that are directly regulated by ANAC102, along with expression analysis, binding peaks, and important citations (Extended Data Figure 28, and 29).

Arnaud D, Deeks MJ, Smirnov N (2023). Organelle-targeted biosensors reveal distinct oxidative events during pattern-triggered immune responses. *Plant Physiol.* 191: 2551-2569.

Orth T, Reumann S, Zhang X, Fan J, Wenzel D, Quan S, Hu J (2007). The PEROXIN11 protein family controls peroxisome proliferation in *Arabidopsis*. *Plant Cell.* 19: 333-350.

Comment 4 by Reviewer #3

4. Figure 7 needs to complement the abscission phenotypes of atg8a, the staining results of DAB and Trypan blue, and the abscission phenotype of atg7 can be included in the supplementary data.

Response

We have added an analysis of the *atg8a* mutant. For this purpose, we obtained the *atg8a* mutant from TAIR and conducted phenotypic analysis. *atg8a* (SALK_012133) harbors a T-DNA within the gene body, resulting in undetectable levels of *ATG8a* transcripts. The *atg8a* mutant has delayed petal abscission,

accompanied by lower ROS accumulation, and fewer dead cells at the petal base than the WT.

In the revised manuscript, we present the *atg8a* characterization in Fig. 7. We describe genotyping of *atg8a* and the *atg8a* and *atg7* mutant phenotypes in the supplementary figures (Extended Data Figures 30–32).

Comment 5 by Reviewer #3

Extended Data Fig. 3 should be annotated with position.

Response

We have made the requested change.

REVIEWERS' COMMENTS

Reviewer #1 (Remarks to the Author):

In the revised version of the manuscript, the authors answered the questions raised by the reviewer. And I have no further questions.

Reviewer #2 (Remarks to the Author):

The authors have addressed the points that I raised in my report. Congratulations for a nice study.

Reviewer #3 (Remarks to the Author):

The author has answered all my questions and I have nothing else to ask. This study innovatively proposed the role of autophagy in the regulation of Arabidopsis petal abscission, expanding the understanding of the process of programmed cell death at postabscission.